# Irrational choices via a curvilinear representational geometry for value

Katarzyna Jurewicz [1,2], Brianna J. Sleezer[3], Priyanka S. Mehta[3,4], Benjamin Y. Hayden [5] & R. Becket Ebitz [1] ✉

We make decisions by comparing values, but it is not yet clear how value is represented in the brain. Many models assume, if only implicitly, that the representational geometry of value is linear. However, in part due to a historical focus on noisy single neurons, rather than neuronal populations, this hypothesis has not been rigorously tested. Here, we examine the representational geometry of value in the ventromedial prefrontal cortex (vmPFC), a part of the brain linked to economic decision-making, in two male rhesus macaques. We find that values are encoded along a curved manifold in vmPFC. This curvilinear geometry predicts a specific pattern of irrational decision-making: that decision-makers will make worse choices when an irrelevant, decoy option is worse in value, compared to when it is better. We observe this type of irrational choices in behavior. Together, these results not only suggest that the representational geometry of value is nonlinear, but that this nonlinearity could impose bounds on rational decision-making.

Converging evidence suggests that we make decisions by estimating and comparing the values of available options. But how are values represented in the brain? Early single-neuron work on value encoding found neurons whose firing rate changed essentially monotonically in proportion to the value of an offer[1]. Since that time, a significant linear relationship between firing rate and value has become the standard test of whether or not a neuron is tuned for value[2–9]. However, some studies have also reported neurons with nonlinear tuning for value in key value-coding regions[8,10,11]. These observations could suggest that value is no different from other continuous, linear variables like speed, contrast, and numerosity, where single neurons can be tuned to prefer specific values of those variables, rather than having uniformly monotonic response profiles[12–17]. In many of these cases, the continuous range of the variable is represented only at the population level through the collective patterns of activity across the population of neurons. However, because nonlinearities in the neural representation of value would make it difficult to make rational economic

decisions[8], the idea that neurons may be non-linearly tuned for value remains controversial.

Part of the reason this controversy persists is that we have historically estimated value tuning via fitting tuning functions to the noisy responses of single neurons. Spiking noise makes it difficult to determine if a neuron truly has a nonlinear response profile, or else just happened to fire more than average for some values during a finite sampling window. It is also not clear whether nonlinear tuning in a small number of single neurons would have any consequence for rational decision-making, provided the representation of value at the neural population level remains essentially linear. Fortunately, recent advances in neural analysis techniques now allow us to look at neuronal populations holistically and probe the representational geometry of value within decision-making circuits with unprecedented resolution[18]. This means that we can finally empirically determine whether the representational geometry of value is indeed linear, in line with the common assumption, or else if it takes on some higher

[1]Department of Neurosciences, Faculté de médecine, and Centre interdisciplinaire de recherche sur le cerveau et l'apprentissage, Université de Montréal, Montréal, QC, Canada. [2]Department of Physiology, Faculty of Medicine and Health Sciences, McGill University, Montréal, QC, Canada. [3]Department of Neuroscience, Center for Magnetic Resonance Research, and Center for Neuroengineering, University of Minnesota, Minneapolis, MN, USA. [4]Psychology Program, Department of Human Behavior, Justice, and Diversity, University of Wisconsin, Superior, Superior, WI, USA. [5]Department of Neurosurgery, Baylor College of Medicine, Houston, TX, USA. ✉e-mail: becket@ebitzlab.com

dimensional geometry, like a curved manifold[19,20] or even a high-dimensional "tangled" manifold[21,22].

What are the functional implications of linear and nonlinear representational geometries of value? Or, in other words, why has the linear hypothesis been so tempting? A linear system is ideally suited for value-based calculations: in linear representation, each unit change in value produces a unit change in neuronal activity. Thus, in principle, a linear system is suitable for generating choices that satisfy axiomatic requirements of rational decisions[8]. However, non-linearity (and irrationality) is ubiquitous in value-based decisions, as we well know from the field of behavioral economics. Many value-based decisions are not actually rational or else are only rational under the assumption that accuracy is bounded by some kind of cognitive or computational limitations[23,24]. This latter idea, known as "bounded rationality"[25,26], is generally linked to constraints on specific cognitive capacities such as working memory or attention, but it is just as plausible that constraints on how the brain can represent value could impose their own bounds on rational decision-making.

Here, we first characterized the representational geometry of value then examined its consequences for rational decision-making. We focused on neurons in the ventromedial prefrontal cortex (vmPFC, area 14). Among value-sensitive regions, the vmPFC is perhaps the most strongly associated with evaluation and choice processes, and in representation of value along the kind of common scale necessary for

economic decisions[10,27–30]. We measured vmPFC responses to the value of offers that were encountered during a menu-search task[11]. This task was ideal for probing the representational geometry of value because offers were unidimensional, lacked any ambiguity or risk, were encountered sequentially (multiple times per trial), and were uniformly distributed across value space. We found that relaxing the assumption that neurons must be linearly tuned for value netted a larger number of value-tuned neurons, some of which had obviously nonlinear tuning functions. At the population level, the representational geometry of value lay along an ordered, but curvilinear manifold in the neural state space, consistent with emerging evidence that curvilinear manifolds may be a ubiquitous feature of population codes, even for linear variables[19,20,31,32]. Because of its curvature, this representational geometry predicted a specific pattern of mistakes—a specific and paradoxical type of irrational choices that we also observed in menu-search task behavior. Together, these results suggest that at least some aspects of bounded rationality may derive from constraints on neural population coding.

## Results

Two rhesus macaques performed a total of 44,335 trials (subject J: 23,826 trials; subject T: 20,509 trials) of a menu-search task (Fig. 1a). Some analyses of this data have been presented previously[11]. All analyses presented here are new. On each trial, a menu of 4 or 7 masked

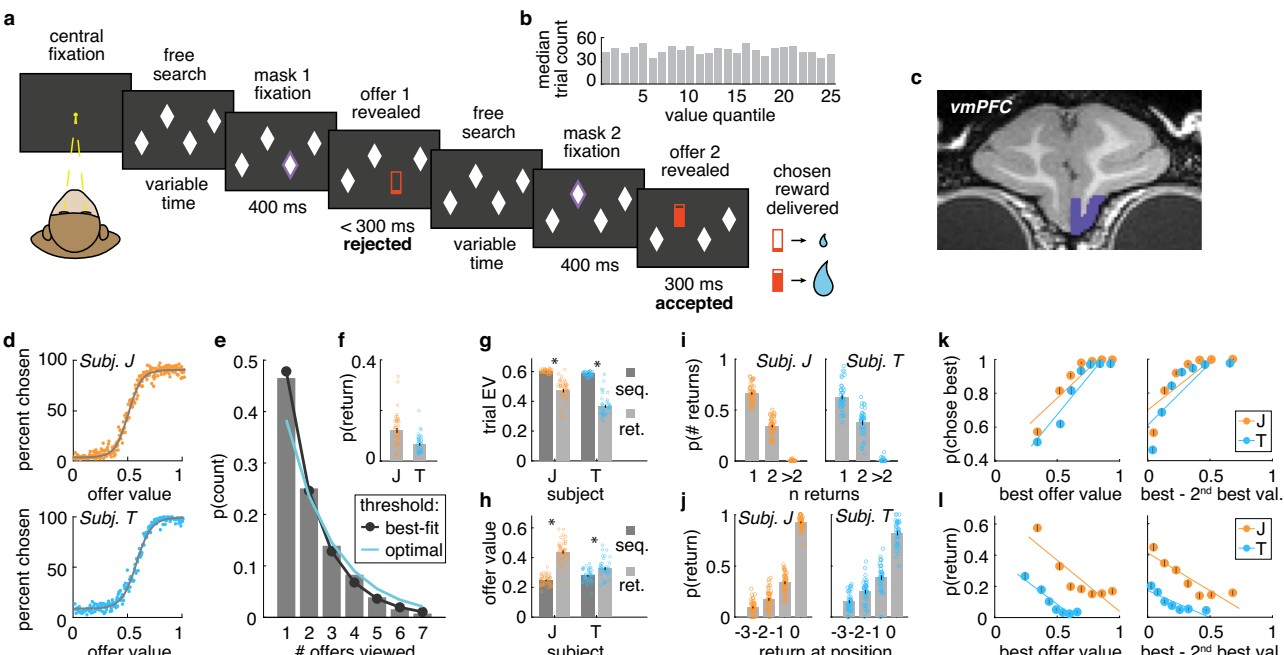

**Fig. 1 | Task design and behavior. a** Subjects chose from menus of 4 or 7 masked offers (4 shown here). Fixating a mask (white diamond) caused it to disappear, revealing a reward cue (red bar). The filled area of the reward cue indicated juice magnitude. The subjects could freely search through the offers in any order. Subjects could accept offers by continuing to fixate them or else could reject an offer and continue to search by saccading away. **b** Offer values were uniformly distributed between 0 and a maximum juice value. **c** Recording sites in vmPFC (area 14). **d** Probability of choosing an offer as a function of offer value. Grayline = logistic fit. Each dot represents an offer value. **e** Distribution of the number of offers viewed per trial. Cyan line = prediction from an optimal compare-to-threshold model. Black line = best fit compare-to-threshold model (maximum likelihood). **f** Probability of returning to a previously seen offer. Each dot represents one session. **g** Expected value of trials which did ("ret." = return) or did not ("seq." = sequential) contain a return to a previous offer, subject J: mean difference = 0.12, 95% CI = [0.11, 0.15], t(88) = 12.01, p < 0.001, two-sample t-test; subject T: mean difference = 0.22, 95% CI = [0.20, 0.24], t(80) = 20.38, p < 0.001, two-sample t-test.

Each dot represents one session. **h** Average value of offers which were returned to ("ret." = return) versus those which were seen only once ("seq." = sequential), subject J: mean difference = −0.19, 95% CI = [−0.21, −0.17], t(88) = −15.92, p < 0.001, two-sample t-test; subject T: mean difference = −0.05, 95% CI = [−0.07, −0.02], t(80) = −4.06, p < 0.001, two-sample t-test. Each dot represents one session. **i** Probability of a given number of returns in a trial, in trials where multiple unique offers were revealed (≥3). Each dot represents one session. **j** Probability of the return in multiple offer trials as a function of the viewing time at which an offer was finally accepted (at position 0). Each dot represents one session. **k** Probability of choosing the best option as a function of value of the best offer revealed within a trial (left) or as a function of the difference in value between the best and the 2nd best offer (right), the two collinear factors influencing overall difficulty of the trial, multiple offer trials, see also Supplementary Fig. S2A. **i** The same as (**k**) but for returning to a previously seen option, see also Supplementary Fig. S2B. Lines = least squares fit. **f–l** Error bars indicate ± standard error of the mean across sessions (SEM), subject J: n sessions = 45, subject T: n sessions = 41, *p < 0.001.

"offers" was presented. The subjects could learn about each offer by fixating it for 400 ms, at which point the mask disappeared and a reward cue was revealed. Offers were illustrated as filled bars, where the filled area indicated the magnitude of juice. Offers were uniformly distributed within the range of 0 and a maximum juice value, which differed between subjects (Fig. 1b). When the offer was revealed, the subjects chose whether to accept it or to reject it and explore the other offers. To accept an offer, the subjects continued to fixate for an additional 300 ms. To reject, they saccaded away. Trials were separated by a 4 s inter-trial interval. During the experiments, we recorded responses of neurons in area 14, a potential macaque homolog of human vmPFC (Fig. 1c).

Both subjects appeared to understand the task. Both chose the best offer they had seen nearly all of the time (subject J: 96% of the time, subject T: 94%) and choice behavior was well-predicted by offer values in general (Fig. 1d; for 4 parameter logistic function [choice by value], both subjects together: slope = 14.46, intercept = 0.52, scale = 0.89, offset = 0.05, $R^2$ = 0.57, n = 135250 offers; similar results in both monkeys individually, Supplementary Table S1). Subjects also evaluated something close to the optimal number of offers per trial. On average, the subjects evaluated 2.13 offers (±0.01 standard error of the mean [SEM] across trials; subject J: 2.09 ± 0.01; subject T: 2.17 ± 0.01) and the number of options viewed appeared geometrically distributed, consistent with a compare-to-threshold process with a threshold at 0.52 of max reward (Fig. 1e; see "Methods" section). A decision-maker implementing the optimal threshold for accepting an offer in this task (i.e., threshold that allows for maximizing the rate of reward) would, on average, evaluate 2.6 offers per trial (optimal threshold = 0.62[11]). Thus, the estimated threshold was slightly lower than strictly optimal.

Importantly, the design of this task allowed subjects to return to re-evaluate a previously revealed offer at any point. Subjects returned at least once in ~10% of trials (Fig. 1f; subject J: 12.11 ± 0.97%; subject T: 6.62 ± 0.55%). Average value of revealed offers (trial expected value, EV) tended to be lower in trials in which returns occurred (return trials) than in strictly sequential trials, suggesting that returns happened when no offer was clearly above threshold or best (Fig. 1g). However, the returns were not indicative of disengagement or accidental repeated selection because they were more likely to return to high-value than low-value offers (Fig. 1h; average value of the offers subjects returned to, subject J: 0.44 ± 0.011; subject T: 0.33 ± 0.010; average value of the remaining offers in the return trials, subject J: 0.25 ± 0.005; subject T: 0.28 ± 0.007). Further, considering only the trials in which subjects revealed multiple unique offers (i.e., at least three), we found that they tended to make one or two returns per trial (Fig. 1i) and that subjects most often returned to an offer to accept it (though the probability of return also increased towards the end of a trial; Fig. 1j). Taken together, the pattern of the subjects' returns suggested a kind of "dithering" in the face of uncertainty: a process of doubling back to reconsider one or two good options prior to committing to one of them.

Because the subjects could return to a previously-viewed offer, their strategy in this task was best described as a mixture of 2 kinds of decision-making processes: a compare-to-threshold process (where each offer was sequentially evaluated against a threshold and accepted immediately if it was clearly above threshold) and a value-comparison process (where newly encountered offers were compared against previously-viewed and still-available offers). In a compare-to-threshold process, the decision is most difficult if values are not clearly above or below the threshold, in particular, if the best option in forward sweep through the offers is still relatively low. In a value-comparison process, the decision is most difficult when good options are close in value and thus harder to discriminate. We found that the subjects were most likely to make errors (i.e. not choose the best offer) when difficulty, understood as either low best value and/or small difference between the best options, was high (Fig. 1k; best offer value effect, subject J: 2.17,

95% CI = [2.02, 2.33], t(44) = 28.56, p < 0.001; subject T: 2.71, 95% CI = [2.55, 2.87], t(40) = 34.23, p < 0.001, logistic regression; best-2nd best offer value effect, subject J: 2.11, 95% CI = [1.95, 2.26], t(44) = 27.48, p < 0.001; subject T: 2.59, 95% CI = [2.45, 2.74], t(40) = 35.51, p < 0.001). Both subjects were also most likely to return to previously-viewed offers when the difficulty was high (Fig. 1l; best value, subject J: −1.65, 95% CI = [−1.87, −1.43], t(44) = −14.97, p < 0.001; subject T: −1.94, 95% CI = [−2.16, −1.72], t(40) = −17.78, p < 0.001; best-2nd best, subject J: −1.33, 95% CI = [−1.53, −1.12], t(44) = −12.99, p < 0.001; subject T: −1.24, 95% CI = [−1.42, −1.05], t(40) = −13.72, p < 0.001). Because the two types of difficulty tended to co-occur (option values tended to be closer when no option was high; r = 0.72; Supplementary Fig. S12), we further verified that both forms of difficulty predicted both errors and returns via regression on the residuals from models that controlled for the other type of difficulty (Supplementary Fig. S2). In sum, both subjects evaluated close to the optimal number of unique offers in each trial, but they did so via a mixed strategy that compared offers both to a fixed threshold and to previous offers encountered within the trial.

## Neuronal activity in vmPFC scales with value

The responses of 122 neurons were recorded from vmPFC (n = 70 in subject J, n = 52 in subject T). To analyze value responses in vmPFC, the trials were broken down into a series of "offer viewing periods": the 500 ms epochs starting 100 ms after the reveal of each offer, to account for sensory processing delays[30]. This epoch was chosen a priori to match the epoch in which the largest number of neurons were modulated by offer value in previous analyses of this dataset[11]. All results were robust to the specific choice of epoch, as described below (Supplementary Fig. S3). The following results were also not due to viewing offers sequentially because all results were also observed just within the first offer viewing period of each trial (Supplementary Fig. S4).

The firing rates of 46/122 (38%) vmPFC neurons were related to offer value (Fig. 2a–d; Supplementary Fig. S1A–C; significantly higher mutual information between the firing rate and value, compared to shuffled value labels; significantly greater proportion than chance, p < 0.001, one-sided binomial test). On average, across all the neurons, we found that increasing value predicted an essentially monotonic increase in the mean neuronal firing rate (Fig. 2e; Supplementary Fig. S3A, S4C; significant main effect of value bin on the firing rate, beta1 = 0.84, 95% CI = [0.71, 0.98], t(23) = 12.50, p < 0.001, $R^2$ = 87.17, AIC = −174.12, n = 25, k = 3, linear function; beta1 = 0.52, 95% CI = [0.01, 1.02], t(22) = 2.01, p = 0.057, beta2 = 0.33, 95% CI = [−0.16, 0.82], t(22) = 1.33, p = 0.198, $R^2$ = 88.12, AIC = −172.12, n = 25, k = 4, quadratic function, AIC weights for linear vs quadratic fit = 0.7311 vs 0.2689, ns.). However, the specific value tuning functions of individual neurons were more heterogeneous than the averaged neuronal profile. Many had some curvature (Fig. 2a; 19/46 [41%] of tuned neurons had value responses better fit by a quadratic function than a linear function, Mandel's fitting test; see "Methods" section; verified with piecewise linear regression, Supplementary Fig. S5). This was not an artifact of the accept/reject decisions in the task: curvilinear turning was still observed for single neurons within accepted and rejected offers (Supplementary Fig. S1).

Because we were largely interested in understanding the population geometry of value coding, we turned to population-level analyses using the whole sample of neurons (i.e. both tuned and untuned; Fig. 3a–d[18]). The majority of the neurons were recorded asynchronously and were combined into pseudopopulations to perform population analyses (see "Methods" section[33–37]). At the population level, we found that there were systematic, structured relationships between the neural responses to different offers. Offers with similar values were represented by similar patterns of neural activity–patterns that were closer together in the neuronal state space–compared to offers with different values (Fig. 3e; Supplementary Fig. S3C, S4I; significant main effect of the difference between two offers on the

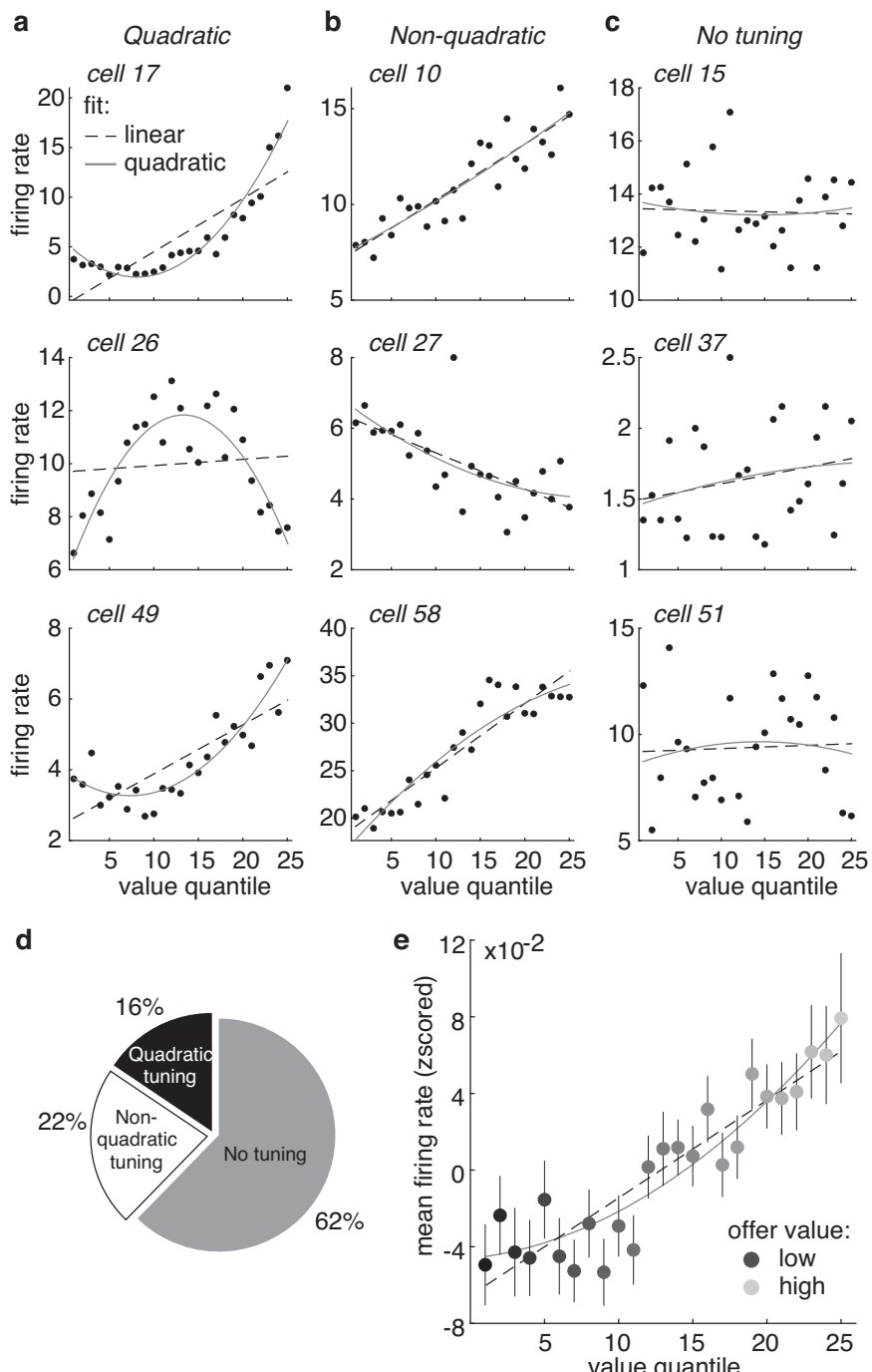

**Fig. 2 | Tuning for value in individual vmPFC neurons. a** The firing rates of three example neurons (rows) that were quadratically tuned for value, plotted as a function of value quantile bins. **b** Same as (**a**) for three example non-quadratically tuned neurons. **c** Same as (**a**) for three example neurons that were not tuned for value. **d** Proportion of all cells (n = 122) within each category. **e** Average firing rates from all 122 neurons, plotted as a function of value quantile. Error bars indicate ± standard error of the mean across neurons (SEM).

representational distance between the offers: beta = 2.32, 95% CI = [2.14, 2.50], t(298) = 25.43, p < 0.001, R² = 0.68, linear regression, n = 300 [all unique pairs of 25 value bins]). Together, these results indicate that offer values were represented in vmPFC in this dataset, both at the level of single neurons and at the level of the population.

### Population activity scales with value along a curvilinear manifold

The vmPFC population represented offer values in a structured way, suggesting that offer value representations were arranged in some

logical order, rather than being "tangled" in a high-dimensional representation in the vmPFC population (Fig. 3b[21,22]). However, there were still at least 2 representational geometries that could produce this structure. For one, offer values could be arranged in a straight line, as a simple, linear sequence of neural states (Fig. 3c). A linear geometry is thought to be important for accurate decoding: for ensuring that downstream structures can correctly and consistently infer which option is the best, no matter the precise set of options the animal is choosing between[8]. Alternatively, offer values could fall along a curved manifold, rather than a straight line (Fig. 3d). Recent

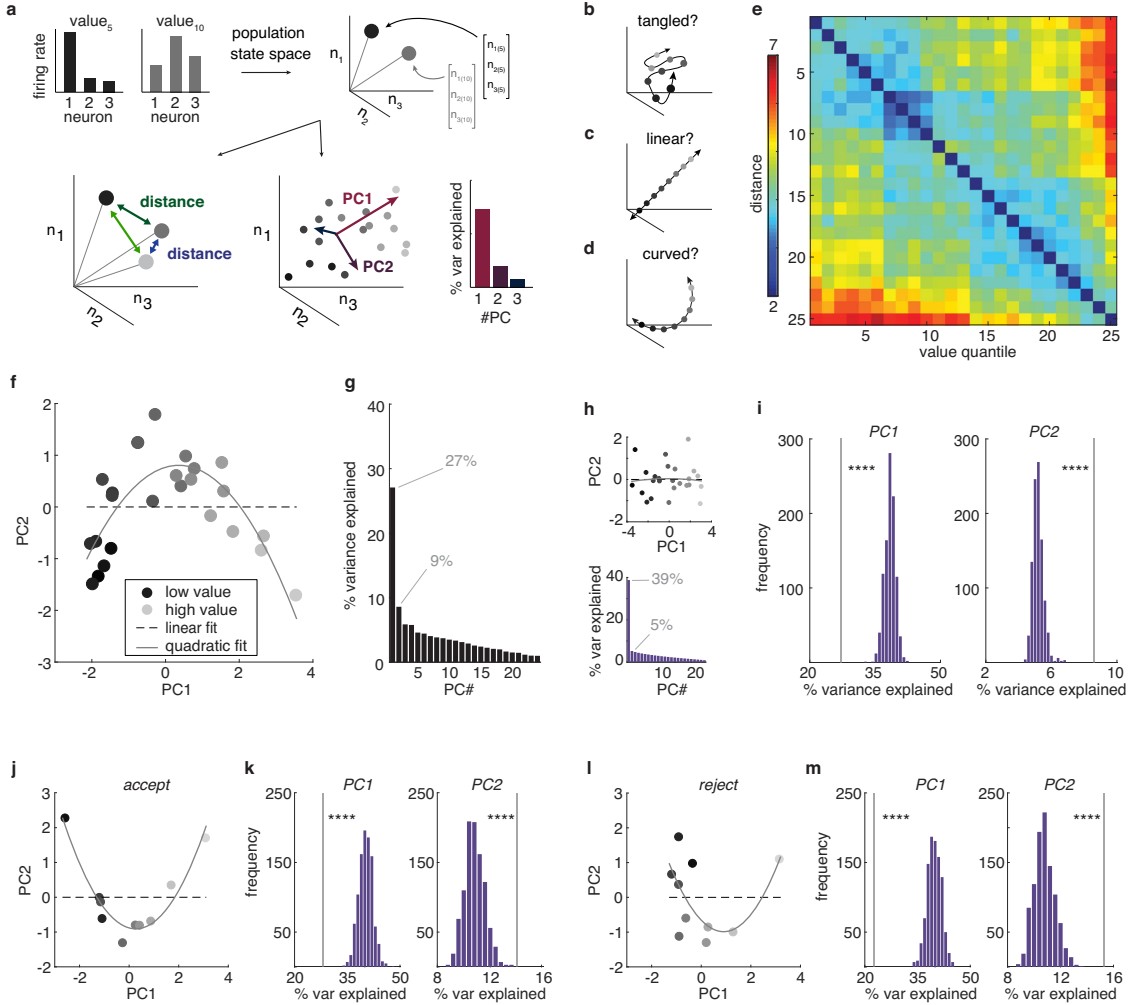

**Fig. 3 | Representational geometry of value is curvilinear at the population level. a** The neuronal population response to an offer value is a pattern of firing rates across neurons (top left). These patterns can also be understood as vectors in a neuron-dimensional space, or, equivalently, as points in the neuronal state space (top right). We probe the representational geometry of these neuronal population patterns through measuring the distance between neural responses (bottom left) or via examining the major axes of co-variability between neurons with principal components analysis (PCA; bottom right). **b**–**d** We considered three hypotheses about how value could be represented at the population level. First, value representations could be "tangled" (**b**): if population value representations are very high dimensional at the population level, nearby values might not even be represented by nearby patterns of activity. Second, value representations could be "linear" (**c**): population value representations could follow a single straight line through the neural state space or, equivalently, occupy a single dimension in the neural state space. Third, value representations could be "curved" (**d**): value representations could be structured–with nearby values represented by nearby patterns of activity–but the population manifold could still occupy more than one dimension. **e** The mean distance between neuronal states corresponding to

different values. **f** The projection of the neural population onto the first 2 principal components (PCs). Shades of gray = value bins from low (light gray) to high (dark gray). Dotted line = best linear fit. Solid line = best quadratic fit. **g** Percent variance explained by each PC. Capturing the variance in a curved function would require more than one PC. **h** Same as (**f**, **g**) for one example linearized dataset (see "Methods" section). **i** A comparison of the variance explained by the first 2 PCs in the real population (vertical line) against bootstrapped distributions of linearized datasets, PC1: 27.08% vs 37.66%, 95% CI = [36.36%, 41.24%], bootstrapped estimate, p < 0.001; PC2: 8.60% vs 5.43%, 95% CI = [4.73%, 5.92%], bootstrapped estimate, p < 0.001. Note that third and higher order PCs also continue to explain more variance in the vmPFC data compared to linearized controls (see "Results" section). **j**, **k** Same as (**f**, **i**), but for accepted offers only, PC1: 27.97% vs 41.29%, 95% CI = [37.0%, 44.78%], bootstrapped estimate, p < 0.001; PC2: 14.04% vs 10.63%, 95% CI = [9.61%, 12.82%], bootstrapped estimate, p = 0.001. **l**, **m** Same as (**f**, **i**), but for rejected offers only, PC1: 22.69% vs 36.86%, 95% CI = [36.23%, 43.73%], bootstrapped estimate, p < 0.001; PC2: 15.27% vs 12.43%, 95% CI = [9.65%, 12.67%], bootstrapped estimate, p < 0.001. n = 121. ****p ≤ 0.001.

studies find that at least some forms of perceptual information may be encoded along curved population manifolds[20], though it is not clear whether reward value might be encoded with this kind of geometry.

One way to arbitrate between the linear hypothesis and the curved hypothesis is to use principal components analysis (PCA). PCA reduces the dimensionality of high-dimensional datasets, like neural data via finding an orthogonal set of linear axes that explain decreasing amounts of variance in the data (principal components; PCs). By projecting neural data onto the first few PCs, we can generate a low-

dimensional intuition for the structure of the high-dimensional population response. Projecting vmPFC population activity onto its first 2 PCs revealed a curvilinear function (Fig. 3f, g; Supplementary Figs. S3D, E, S4D, E). The shape of offer values in the reduced-dimensional-space was better described by a curved, quadratic function than a linear function (linear function: AIC = 74.21, AICc = 74.38, BIC = 77.86, n = 25, k = 3; quadratic function: AIC = 49.16, AICc = 49.71, BIC = 54.04, n = 25, k = 4; all AIC, AICc and BIC weights for the linear function <0.001). Curvature was not apparent when offer representations were first linearized, indicating that this was not some artifact of

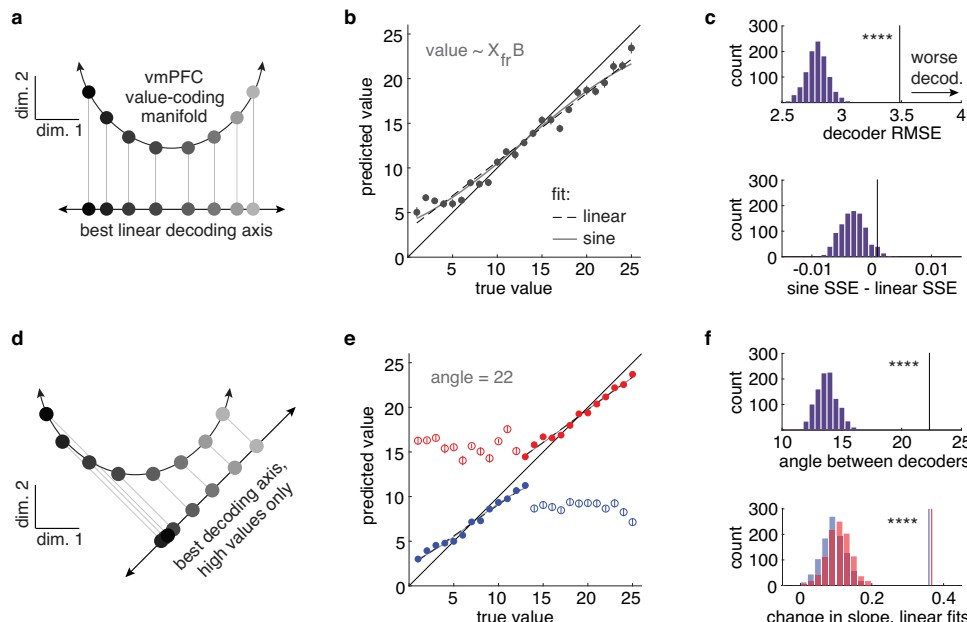

**Fig. 4 | Curvilinear manifolds predict systematic biases in decoding. a** A curvilinear geometry would produce biases in the accuracy of a linear decoder. This cartoon illustrates how the projection of an arc onto a line compresses the values at the tails of the range. **b** A linear decoder trained on the vmPFC population and used to predict value. Note that predicted low values are higher than the true values and predicted high values are lower. **c** (top) Comparing the accuracy of a decoder trained on the real vmPFC data (vertical line) with decoders trained on linearized control populations (purple bars), 3.49 vs 2.82, 95% CI = [2.64, 2.99], bootstrapped test, p < 0.001. RMSE: root mean square error, higher = less accurate. **c** (bottom) Points projected from an arc onto a line, as visualized in (**a**), would have residual errors that follow a sine function. Difference in the quality of a fit of a linear and sine function to the residuals from vmPFC (vertical line) and the linearized controls (purple bars), 0.001 vs −0.003, 95% CI = [−0.006, 0.002], bootstrapped test, p = 0.057. SSE - sum of squared estimate of errors. **d** Another bias predicted by a

curved–but not by a linear–manifold affects out-of-range observations. A decoder trained on a portion of the curved function would not make accurate predictions about the values in the other portion of the curve. **e** Decoders trained on the population response to one-half of the values (filled circles) and used to predict values outside of this range (open circles). Red = trained on high values; blue = trained on low values. **f** (top) The angle between the high- and low-value-trained decoders in real vmPFC population (vertical line) and in linearized controls (purple bars), 22.33 vs 13.97, 95% CI = [12.27, 15.76], bootstrapped test, p < 0.001. **f** (bottom) The change in slope in (**e**) between within-range and out-of-range values for each decoder (red = trained on high values; blue = trained on low), high-trained: 0.37 vs 0.12, 95% CI = [0.06, 0.18], bootstrapped test, p < 0.001; low-trained: 0.36 vs 0.1, 95% CI = [0.04, 0.17], bootstrapped test, p < 0.001. Vertical lines correspond to the vmPFC data and distributions to the control populations. Error bars indicate ± standard error of the mean across neurons (SEM), n = 121. ****p < 0.001.

data processing (see "Methods" section; Fig. 3h; linear function: AIC = 61.38, AICc = 61.56, BIC = 65.04, n = 25, k = 3; quadratic function: AIC = 63.43, AICc = 63.97, BIC = 68.30, n = 25, k = 4; AIC and AICc weights for the quadratic function >0.3, BIC = 0.2[20]).

PCA can also be used to look at curvature in the non-reduced neuronal state space: via asking how succinctly the population response can be approximated by a set of linear axes. If offer values were represented linearly, then we should be able to capture nearly all of the variance between offers with a single PC. However, the variance in a curved function would necessarily span 2 or more dimensions: one to describe the axis spanning the arms of the curved function and one to describe the axis of curvature. (Note that curved functions can occupy more than 2 dimensions in which case they will have more than one axis of curvature; Supplementary Fig. S6) Therefore, to determine if the data was more curved than we would expect from noise, we compared the number of PCs needed to explain the variance in the vmPFC data against linearized control populations (see "Methods" section). The first PC explained significantly less variance in the vmPFC data than in linearized data (Fig. 3i; Supplementary Figs. S3F, S4F; see "Methods" section). Higher order PCs explained significantly more (PCs 2–4, p < 0.001; PCs 5–14, p < 0.01; PCs 15–24, ns.). This was not an artifact of the fact that the task design required subjects to either accept or reject an offer as we observed the same pattern within both accept (Fig. 3j, k, PC1, p < 0.001, PC2, p = 0.001; PCs 3–9, p < 0.02) and reject (Fig. 3l, m; PCs 1–8, p < 0.001, PC9 ns.) decisions separately. Together, these results suggested that offers were represented curvilinearly in the vmPFC population.

## Warped decoding from the curvilinear manifold

If offer values are encoded curvilinearly in vmPFC, it would affect our ability to accurately decode value. Linear decoders are ubiquitous in population analyses[18,38]. They are computationally tractable, approximate most of the variance in many curved functions, and mirror the linear weighted sum over a population that a real downstream structure would perform[39,40]. However, even in the best linear approximation to a curved manifold, decoding accuracy would be warped (Fig. 4a). If we compared, against their true value, values that had been encoded along a curve, then decoded from a linear projection, we would see a subtle, but systematic compression at the tails. High values would appear lower than they actually are, and low values would appear higher than they actually are. To determine if decoding accuracy was compromised in this way, we compared the predictions of a linear decoder with the true underlying values (Fig. 4b; see "Methods" section). Although inaccuracies in model fitting mean that one might see compression at the extreme values even if the underlying value-coding axis was linear, the best decoder on the real vmPFC data was significantly less accurate than decoders trained on linearized control populations (Fig. 4c; larger root mean squared error [RMSE]).

Because the projection of an arc onto a line is a sine function, if values were encoded along a curved manifold, we might expect our residual decoding errors to follow a sine function (Fig. 4a). In contrast, if compression is just due to inaccuracies in model fitting, residual errors should still be linear (indeed they were in linearized populations: linear RMSE = 0.022; sine RMSE: 0.024; the linear function fit better in 897/1000 bootstrapped samples). In the vmPFC data, the

shape of residual errors was better described by a sine function than a linear function, though this was not significantly outside the distribution of linearized populations (Fig. 4c). However, an arc might be a poor approximation to the quadratic functions that appeared to fit the population manifold in 2 dimensions (Fig. 3f). Although we found no closed form solution for the projection of a quadratic function onto a line, the simulation suggested that we might expect the distribution of residuals in that case to asymptote vertically, rather than horizontally, consistent with what visual inspection suggested might be true for high values (Fig. 4b).

If values were represented linearly, we would be able to extrapolate a decoder trained on any portion of the manifold to predict the ordering of values on another. Conversely, in a curvilinear manifold, a decoder trained on one "arm" should offer little information at all about the values on the other (Fig. 4d). Therefore, we next split the values in half and trained 2 separate linear decoders–one on the high values and one on the low values. We then used these decoders to predict the held-out low or high values, respectively. These decoders were significantly less accurate in the vmPFC data than in the linearized control populations (Fig. 4e; high-trained, held-out low RMSE, vmPFC data = 9.94, linearized data = 7.29, 95% CI = [6.5, 8.06], bootstrapped test, $p < 0.001$; low-trained, held-out high RMSE, vmPFC data = 11.50, linearized data = 6.64, 95% CI = [5.86, 7.44]; $p < 0.001$). The best decoding axis for the high values and low values differed more in the vmPFC data, compared to the linearized data (Fig. 4f). Further, while there was little change in the relationship between true and predicted values across test and train in linearized data, this was not the case in the vmPFC population (Fig. 4f). In fact, there was essentially no relationship between the decoder's predicted value and the true value for out-of-range values in the vmPFC data (high-trained $r = -0.01$; low-trained $r = -0.0002$). In sum, we found systematic inaccuracies in decoding values from the vmPFC population that were consistent with the idea that values were encoded along a curved, rather than linear manifold.

## Irrational choices from a curvilinear manifold

If a downstream neuron was taking a weighted sum of the activity in vmPFC neurons, it would be acting as a linear decoder: its input would be some linear projection of the curved vmPFC manifold. However, linear approximations to curved functions are only accurate locally: they are perfect approximations in the limit of instantaneous segments, but increasingly inaccurate over wider portions of the curve. This implies that there is an upper bound on how accurately value can be decoded from vmPFC by a downstream neuron and that this upper bound would change with the range of values that must be decoded. For narrow ranges, decoding would be fairly accurate, and nearby values easy to discriminate (Fig. 5a). However, as the range of values increases, even the best possible decoder would begin to produce systematic errors, like confusing nearby offers or confusing the precise value of near-threshold offers.

In the context of economic decision-making, one systematic error we would expect from a curved manifold is a specific violation of one of the principles of rational choice theory: the independence of irrelevant alternatives axiom[41–43]. Rational decision-making requires that choices should not be affected by the availability of irrelevant, "decoy" offers whose value is low enough that they are unlikely to ever be chosen. However, the curved manifold implies that especially low-value offers should actually interfere with our ability to make good decisions because they decrease the upper bound on the accuracy of decoding higher-value offers from vmPFC activity (Fig. 5a).

We began to test the idea that decoy offers should compromise value-based decision-making through decoding different sets of offers from vmPFC activity. We found that curvature in the value-encoding manifold was sufficient to produce strong decoy effects in decoding accuracy (Fig. 5b). Linearized populations also produced a weaker but

positive decoy effect, suggesting that the linearized population still maintained a level of curvature capable of affecting the readout, likely due to the Poisson spiking close to the noise floor[20]. This result was not an artifact of considering some especially wide range of values: curvature in value-encoding manifold was sufficient to produce strong effects on decoding accuracy when we only considered a narrower range of values (Supplementary Fig. S7). To test the role of curvature in producing the decoy effect we additionally simulated 50 curved and linear populations with slightly elevated baseline firing rates to remove them from the noise floor (Supplementary Figs. S8, S9). Here, we found that curvature in the manifold was necessary to produce decoy effects in decoding accuracy, as the decoy effect was absent in simulated linear populations (Fig. 5c). In short, a curved value manifold predicts a paradoxical behavioral phenomenon where high decoy offers would make it easier to discern which offer in a given set is best. (Note that this is the *inverse* of what a divisive normalization account would predict, where low-value [not high-value] decoys would improve decoding good offer values because lower decoys reduce the magnitude of the divisor component[44,45]).

In addition to being better at choosing the best option as it got larger with respect to the threshold and/or in comparison to the second-best option, the subjects were also better at choosing the best option as the value of the worst, decoy option in each set got higher (Fig. 5d, e; GLM included terms for choosing the best option by best value, best-2nd best value, decoy value, and pairwise interactions with decoy value, similar results for individual subjects, and different parameterizations of the GLM, Supplementary Tables S2 and S3, Supplementary Fig. S11). While accuracy for very high best offer values approached the ceiling, a strong effect of decoy value was observed for lower best offer values and best offer values that were close to the 2nd best offers (Fig. 5f; both subjects: decoy by best interaction, mean slope = 0.27, 95% CI = [0.26, 0.29], $t(85) = 33.94$, $p < 0.001$; decoy by best-2nd best interaction, mean slope = 0.35, 95% CI = [0.32, 0.38], $t(85) = 24.04$, $p < 0.001$; similar results for individual subjects, Supplementary Table S2, Supplementary Fig. S10A–C).

Errors of reward maximization are only one of the consequences of uncertainty about the best option in the task. As described in Fig. 1, the monkeys were also more likely to return to previous options on trials where the best decision was uncertain. Therefore, we reasoned that any effect of decoys on the discriminability of good options or of good options from threshold might also result in an increase in returns. As predicted, subjects were less likely to dither between options (i.e. they made fewer returns) when the value of the worst option in the set got higher (Fig. 5g–i). This was true even after taking into account the best offer value and the difference between the best and 2nd best offers (same GLM as for accuracy, similar results for individual subjects, Supplementary Table S2). Again, the decoy effect was most pronounced for lower best offer values and best offer values close to the 2nd best offers (Fig. 5i; both subjects: decoy by best interaction, mean slope = −0.43, 95% CI = [−0.48, −0.37], $t(85) = −15.83$, $p < 0.001$; decoy by best-2nd best interaction, mean slope = −0.18, 95% CI = [−0.20, −0.15], $t(85) = −16.10$, $p < 0.001$; similar results for individual subjects, Supplementary Table S2; Supplementary Fig. S10D–F). Together, these results converge to suggest that the subjects were more uncertain about which offer to choose when irrelevant decoy offer values were lower, tracking the upper bound on decoding accuracy predicted by the curved manifold.

Nonetheless, because the task was not designed to test for decoy effects, we considered the possibility that the decoy effects were an artifact of some confounding variable that might be correlated with decoy value. We found decoy value consistently improved choice accuracy and decreased the probability of returns in a way that could not be explained by any differences in set size, decoy order effects, decoy recency effects, distraction by the decoy, or some interaction between accuracy and returns. First, while subjects were more likely

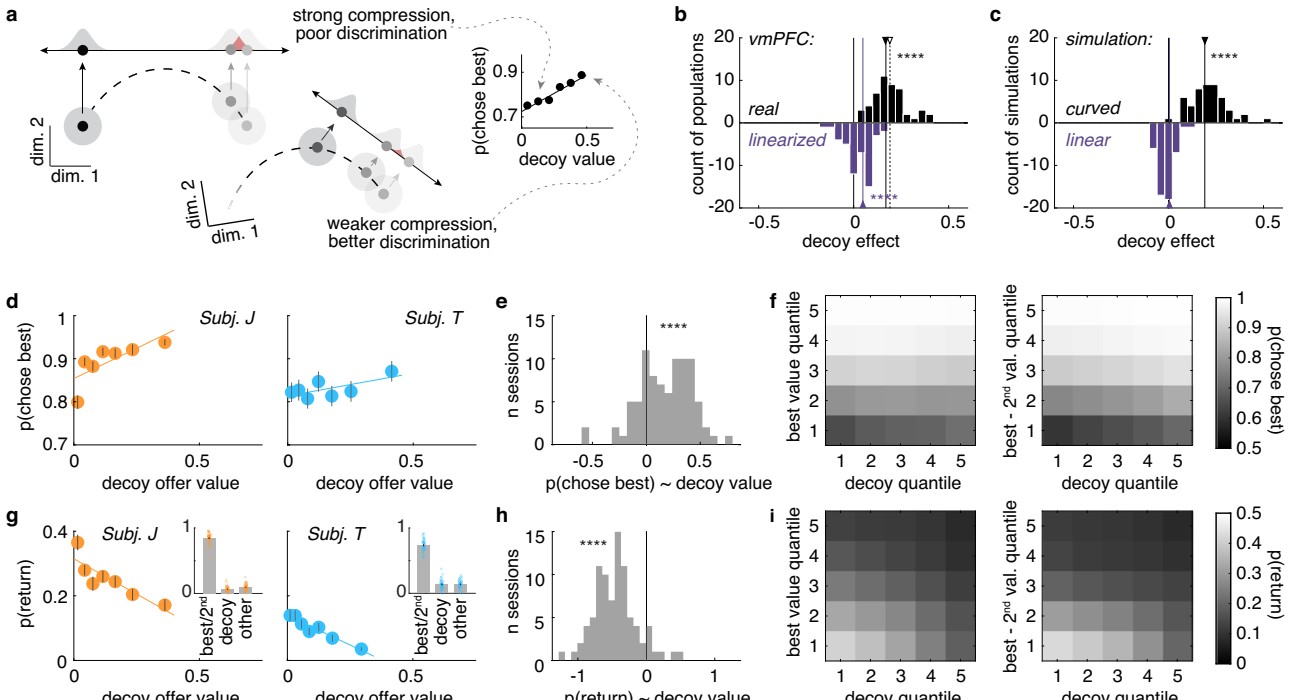

**Fig. 5 | Curvilinear manifolds suggest a specific pattern of irrational decisions.**
**a** A cartoon illustrating how the ability to discriminate good offers could depend on the value of a worst, "decoy" offer in a curved manifold. The linear decoder represents an upper bound on the accuracy of a downstream neuron's readout from vmPFC. The inset graph shows how the probability of choosing the best option changes as a function of decoy offer value in an example vmPFC pseudo-population; the slope here is the "decoy effect": the effect of the worst offer's value on the accuracy of the choices (marked in [**b**] with an open arrow). **b** Distributions of decoy effect slopes from vmPFC pseudopopulations (black) and their linearized version (purple), vmPFC: mean slope of the decoy effect = 0.17, 95% CI = [0.14, 0.20], t(49) = 13.42, p < 0.001; linearized population: mean slope of the decoy effect = 0.05, 95% CI = [0.03, 0.07], t(49) = 4.89, p < 0.001, one-sample t-test from 0. Filled arrows = means, open arrow = the example effect illustrated in (**a**). **c** Same as (**b**) but the distributions of decoy effects were obtained from simulated populations that had either a curved (black) or linear (purple) value manifold; curved

population: mean slope of the decoy effect = 0.19, 95% CI = [0.16, 0.22], t(49) = 13.91, p < 0.001; linear population: mean slope of the decoy effect = 0.005, 95% CI = [−0.007, 0.017], t(49) = 0.79, p = 0.436, one-sample t-test from 0.
**d** Probability of choosing the best offer seen as a function of the worst offer seen (decoy offer quantile). Lines = least squares fit. **e** Distribution of decoy effect slopes across sessions, mean decoy effect slope = 0.17, 95% CI = [0.12, 0.23], t(85) = 6.50, p < 0.001, one-sample t-test from 0. **f** Probability of choosing the best option as a function of the decoy value and other variables that affect accuracy: (left) the best value seen; (right) the difference between the best and 2nd best option. **g–i** Same as (**d–f**), but for the probability of returning to a previously seen offer, mean decoy effect slope = −0.45, 95% CI = [−0.52, −0.38], t(85) = −13.42, p < 0.001, one-sample t-test from 0. Insets in (**g**) show which options tended to be the target of returns. Each dot represents one session. Error bars indicate ± standard error across sessions (SEM), subject J: n sessions = 45, subject T: n sessions = 41. These may be smaller than the symbols. ****p < 0.001.

to see low-value decoys when they viewed more offers in a trial (sig. correlation, r = −0.33, p < 0.001), the decoy value effect was present also after accounting for the number of offers revealed (Supplementary Fig. S11A–C). Second, decoy value influenced choices and returns across decoy's position in the trial sequence (Supplementary Fig. S11D–F) and its recency to the time of the choice (Supplementary Fig. S11G–I). Third, neither the accuracy nor return effects were due to distraction by the worst value. The majority of the returns went to the best or 2nd best option (Fig. 5g, insets; subject J: 84 ± 6%, subject T: 72 ± 2%), with no more attention paid to the worst option than any other options (subject J: 6.42 ± 0.61% vs 9.44 ± 0.73%, mean difference other - worst = 3.02%, 95% CI = [1.70%, 4.34%], t(44) = 4.60, p < 0.001, subject T: 13.42 ± 1.17% vs 13.57 ± 1.04%, mean difference other - worst = 0.15%, 95% CI = [−3.15%, 3.45%], t(40) = 0.09, p = 0.927) and the likelihood that the subjects chose the worst option was negligible (subject J: 2.09 ± 0.16% subject T: 3.65 ± 0.20%). Finally, while choice accuracy and the probability of returns were related (Supplementary Fig. S12), returns did not explain the errors of reward maximization. Decoy effect was still present in choice accuracy within only those trials that did not contain any returns (both subjects: mean slope of decoy effect = 0.05, 95% CI = [0.00, 0.10], t(85) = 2.04, p < 0.044; decoy by best interaction, mean slope = 0.20, 95% CI = [0.18, 0.22], t(85) = 18.58, p < 0.001; decoy by best-2nd best

interaction, mean slope = 0.23, 95% CI = [0.20, 0.27], t(85) = 13.23, p < 0.001 [choosing the best option by best value, best-2nd best value, decoy value, and pairwise interactions with decoy value]). Together, these results suggest that errors and returns both arose when the choices were uncertain and constitute converging pieces of evidence that low-value decoys warp the discriminability of offers.

Again, these effects did not appear to be the simple consequence of divisive normalization. The curvature in single-cell and population data is similar when we only consider the first offer viewing period when the context of other offers in the trial is not yet present (Supplementary Fig. S4). We found some evidence for divisive normalization when considering sequential effects in this dataset (Supplementary Fig. S13): namely, in trials in which the first offer viewed was higher than average, we found a shift in the representation of low-value options such that they would be decoded as slightly higher-value than they were. Because value normalization is linked with the inverse of the decoy effects we report here (i.e. higher decoy values causing worse performance, not better performance), normalization effects would only cause us to underestimate the magnitude of the effect of the curved value manifold on decision-making. In sum, what should have been an irrelevant decoy–the worst option in the set–interfered with decision-making the most when it was furthest

from the best available options, exactly as we would expect if values were decoded from a curved manifold.

## Discussion

Value is the key variable of economic decision-making, so its representation in the cortex likely has consequences for value-based decisions. In vmPFC, a region causally implicated in value-based decision-making[46–48], we found that the average firing rates across neurons scaled positively and monotonically with value. However, many individual neurons were reliably, but non-linearly tuned for value. At the population level, the patterns of neuronal activity that encoded value traced a curvilinear manifold, rather than a straight line. This curvilinear geometry could explain why not every vmPFC study finds robust value tuning[49,50]. Random samples of neurons are low-dimensional projections of the underlying population and a curvilinear value manifold would have many low-dimensional projections that would lack even monotonic tuning with respect to value[20]. The neuronal code for value has long been assumed to be linear and at least quasi-linear tuning for value has been reported in single neurons in many brain regions[2–9]. However, our results reinforce the idea that single neurons can be meaningfully tuned for value without being linearly tuned for value[10,11]. They also suggest that the population-level representational geometry of value can be strongly nonlinear, even when the average response across neurons is not.

If a set of option values is represented linearly, it is trivial for a downstream region to decode these values in a way that respects fundamental axioms of rational value-based decision-making, like transitivity, completeness, and the independence of irrelevant alternatives. The last of these, independence of irrelevant alternatives, is the dictum that decisions between good alternatives should be unaffected by the value of low-value, decoy options[41–43]. However, if the representation of value were curved, decoys would affect decision-making. Because curvature both (1) warps the representation of values, and (2) scales positively with the range of values, the lowest-value decoys would introduce a systematic bound on the upper limit of decoding accuracy for choice-relevant, high-value options. Indeed, we found that subjects exhibited exactly the pattern of choices predicted by their curvilinear manifold: the lower the value of the decoy option in the set, the less accurate the subjects were at choosing among better offers. This result resonates with a broad literature showing that violations of the independence of irrelevant alternative axioms are surprisingly common in real-world decision-makers[45,51–57]. However, it is important to note that high-value decoys can sometimes compromise decision-making more than low-value decoys[45,57]. This may occur because divisive normalization effects dominate when offers are presented simultaneously, rather than sequentially[55]. Future work is needed to fully understand the consequences of curvilinear value encoding for rational decision-making.

The links we draw between the curved value-coding manifold and irrational decision-making may seem predicated on some provocative premises. The first of these is the notion that a downstream region would decode value in a way that is essentially linear. There is precedent for this view. Linear decoders parallel the weighted sum over neurons that a downstream neuron would receive as input[39,40,58] and both neuronal activity[39,59,60] and behavior[19,61] are often well-described as a linear decoding of some more complex pattern of brain activity[19]. Further, though this study cannot say what decoding scheme might be used by any downstream region(s), linear functions offer a good approximation to many nonlinear functions, so our approach may make reasonable predictions even if the true decoding scheme is not strictly linear. The second premise that our hypothesis may appear to depend on is the idea that a different downstream neuron would be responsible for decoding different sets of options. That is, one could imagine that a downstream region might flexibly alter its decoding strategy in order to maximize information about the currently available set. Given that we know that neural representations are constantly evolving, both spontaneously[62,63] and according to internal states, task contexts, and value ranges[4,19,64–69], this does not seem like an unreasonable idea. However, it is worth clarifying that our argument is not predicated on flexible decoding. Instead, we argue only that the curved manifold creates a bottleneck in information transmission: an upper bound on the decoding accuracy possible in all of the neurons downstream from vmPFC. We struggle to see how it would be possible to compensate for an information bottleneck like this, even allowing nonlinearities and complex network effects, but future work is certainly needed to determine how regions downstream from vmPFC decode from this curved manifold.

There is precedent for the idea that single neurons have flexible, context-dependent value representations in the literature on divisive normalization[44,45,55] and range adaptation[4,64,65]. In fact, one might even wonder if the decoy effects here could be due to set-dependent changes in value encoding in single neurons rather than the nonlinear value manifold in vmPFC. However, context or set had only minimal effects on the vmPFC manifold that would have only made the decoy effects weaker, not stronger. Further, neither form of flexible encoding seems like a sufficient alternative explanation for the effects reported here. First, while value normalization can certainly cause interference from low values, it is typically linked with the inverse of these decoy effects: worse performance with higher decoy values, not better performance[45]. Normalization-like effects are also more typical of task designs when options are presented simultaneously, rather than sequentially, perhaps because divisive normalization is related to competition for visual attention[70]. Second, while range adaptation could produce decoy effects with the same sign as those reported here, range adaptation effects typically take place at longer timescales (across blocks or tasks rather than within trials[4,64,65]). Significant adaptation would have also compromised the ability to compare sequentially viewed offers against a stable threshold, but the subjects reliably accepted offers that were above a threshold here. Intriguingly, a curved manifold in vmPFC—a region upstream from the orbitofrontal regions in which range adaptation is maximal[71] could actually be part of the as-yet-unknown mechanism(s) *for* range adaptation. That is, perhaps downstream neurons range-adapt in part because doing so maximizes the information they can decode from nonlinear, curved representational structures. Simultaneous population recordings in vmPFC and orbitofrontal range-adapting neurons might be the key to addressing this question.

In a fixed, quasi-linear decoding from a curved manifold, the separation between nearby values is maximal at the center: at precisely the values that are closest to the accept/reject threshold the animals adopted here. In fact, because the difficulty of the accept/reject discrimination is aligned with the axis of curvature, another possible interpretation of this result is that vmPFC encodes difficulty as well as value. However, we would argue that it is more parsimonious to imagine that vmPFC "encodes difficulty" as a byproduct of the inherent curvature in the representation of value. Curvature emerges naturally from fundamental constraints on the dynamic range of neuronal firing rates (i.e. the lower bound at zero and an upper bound at the neuron's metabolic limits[20]) and appears even in tasks where there is no natural alignment with any difficulty axis[19,31]. Further, even when the axis of curvature does align with task difficulty, position along the "difficulty" axis does not appear to predict the experienced difficulty of the animals' decisions[20]. However, just because information is not used for a specific judgment does not mean that it is not used and future work is needed to determine if the apex of the curvature is sensitive to changes in threshold or else if other neural correlates of decision difficulty could be an artifact of curved value representations[72,73].

The idea that curvature can exist even in a circumstance where it compromises behavior lends credence to the notion that curvature is a ubiquitous feature of the population encoding of many variables,

including linear ones. This study stands alongside growing evidence for curved representational geometries across several brain structures for several linear variables, including those underlying perceptual decisions[20], motor control[31,32], and interval reproduction[19]. Curvature could be a powerful way to maximize information coding within the bounded dynamic range of neuronal firing rates[20]. While linearity might be essential for rational decision-making, a linear information-coding manifold in a bounded space represents a significant compromise in the discriminability of states along its length compared to coding manifolds that have more dimensions[20–22,74]. As the curvature of information-coding manifolds expands into more dimensions, the number of representations that can be discriminably encoded along that manifold will also increase[18]. However, future work is needed to determine whether the curvilinear value manifold has some functional benefits or if it exists as a byproduct of constraints on neuronal firing rates. In either case, the curvilinear geometry of value represents a constraint in the hardware used for value-based decision-making that could explain some types of seemingly irrational decisions.

## Methods
All procedures were designed and conducted in compliance with the Public Health Service's Guide for the Care and Use of Animals and approved by the University Committee on Animal Resources at the University of Rochester. Subjects were two male rhesus macaques (*Macaca mulatta*: subject J age 10 years; subject T age 5 years). Although our ethical mandate to minimize the number of animals meant that sex was not considered in the design of this specific study, future work using the same methods in female macaques would be invaluable. Initial training consisted of habituating animals to laboratory conditions, to head restraint, and then to perform oculomotor tasks for liquid reward. Standard surgical techniques were used to implant a small prosthesis for holding the head and Cilux recording chambers (Crist Instruments) over the vmPFC. Position was verified by magnetic resonance imaging with the aid of a Brainsight system (Rogue Research Inc.). Anesthesia was induced with 5 mg/kg ketamine and 0.02 mg/kg buprenorphine, and it was maintained using gaseous isoflorane at 2%. After all procedures, animals received appropriate analgesics and antibiotics: acutely immediately following surgery and in the recovery treatment over the next 14 days (50 mg/kg dose of cefazolin). Throughout all sessions, the chamber was kept sterile with regular washes and sealed with sterile caps. Previous analyses of these data have been included in earlier work[11], together with subjects' previous training history; all analyses presented here are new. Data were recorded over 45 separate sessions in Subject J and 41 sessions in Subject T. This allowed for collecting the recordings from 122 neurons. Sample size was determined based on previous publications that obtained reliable single-neuron and population effects on a similar number of cells (e.g.[20]). One neuron was excluded from population analyses due to lack of spiking activity in the selected time window.

### Recording sites
We approached vmPFC through a standard recording grid (Crist Instruments), guided by a Brainsight system and structural magnetic resonance images taken before the experiment. Neuroimaging was performed at the Rochester Center for Brain Imaging, on a Siemens 3 T MAGNETOM Trio Tim using 0.5 mm voxels. The accuracy of the Brainsight guidance was confirmed by listening for characteristic differences between white and gray matter during electrode penetrations. Gray-white matter transitions occurred at the penetration depths predicted by the Brainsight system in all cases. We defined vmPFC according to the Paxinos atlas[75]. This meant we recorded from a region of interest lying within the coronal planes situated between 42 and 31 mm rostral to interaural plane, the horizontal planes situated between 0 and 7 mm from the brain's ventral surface, and the sagittal

planes between 0 and 7 mm from the medial wall. This region is within the boundaries of Area 14 according to the atlas.

### Electrophysiological techniques, eye tracking, and reward delivery
A microdrive (NAN instruments) was used to lower single electrodes (Frederick Haer & Co., impedance range 0.7–5.5 MU) until waveforms of between one and four neurons were isolated. Individual action potentials were isolated on a Plexon system (Plexon, Inc.). We only selected neurons based on their isolation quality; never based on task-related response properties. All collected neurons for which we managed to obtain at least 300 trials were analyzed. In practice, 86% of neurons had over 500 trials (this was our recording target each day).

An infrared eye-monitoring camera system (SR Research) sampled eye position at 1000 Hz, and a computer running Matlab (Mathworks) with Psychtoolbox and Eyelink Toolbox controlled the task presentation. Visual stimuli were colored diamonds and rectangles on a computer monitor placed 60 cm from the animal and centered on its eyes. We used a solenoid valve to control the duration of juice delivery, and established and confirmed the relationship between solenoid open time and juice volume before, during, and after recording.

### Experimental design
Subjects performed a menu-search task (Fig. 1a). To begin each trial, the animal fixated on a central dot (50 ms), after which either four or seven white diamonds ("offers") appeared in randomly selected, non-overlapping positions on the screen. The number of offers per trial was either 4 or 7, and was chosen at random on each trial. Continuous fixation on one diamond for 400 ms caused it to disappear and reveal a reward offer. Offers were orange bars, partially filled in to indicate the value of the riskless offered reward. The percentage of the offer bar that was filled in corresponded to the offer value in terms of percent of the maximum value possible per offer (e.g., an offer bar that was 10% orange and 90% black would indicate an offer worth 10% of the maximum value; 20 uL for subject T and 23 uL for subject J). Continuous offer values were discretized by the pixel resolution of the display. Reward values for each offer were generated randomly from a uniform continuous distribution ranging from 0% to 100% of the maximum possible reward value for the individual subject (although continuously varying rewards were generated, the need to represent offers in pixels did discretize offers into approximately ~200 steps).

The subject could freely search through the diamonds in any order and could accept any offer. Acceptance led to the end of the trial; rejection led to a return to the initial state (viewing an array of diamonds). To accept the offered reward, the animal had to maintain fixation on the offer for 300 ms, after which the screen would go black and the offered amount of liquid reward would be delivered immediately. Thus, selecting a given offer required 700 ms: 400 to unmask it and an additional 300 to obtain it. If the subject broke fixation on the reward stimulus at any point between 0 and 300 ms from the initial reveal, the reward stimulus would disappear and the diamond would return in its place (a "rejection" of the offer). The subject could then resume freely inspecting other offers. There was no limit to how many offers a subject could inspect, nor to how many times a subject could return to re-inspect a particular offer. The trial only ended (and a liquid reward was only delivered) after the subject accepted an offer. Reward delivery was followed by a 4 s inter-trial interval.

### General data analysis techniques
Data were analyzed with custom software, written in Matlab. Neural activity was analyzed in the fixed 500 ms epoch beginning 100 ms after the offer was revealed. This epoch was chosen a priori, matching the previous publication[11]. To verify that the results are not primarily driven by the post-decision factors, additional analysis (analogous to population analyses in Figs. 2 and 3) was performed for the epoch

beginning 100 ms and ending 300 ms after the offer was revealed (i.e. the epoch beginning after vmPFC neurons responded to the offer reveal, but before any offer was accepted; Supplementary Fig. S3). All comparisons using t-test were two-tailed. No corrections for multiple comparisons were necessary because the tests were independent. Formal analyses of the normality of distributions were not performed.

## Analysis of choice behavior

The menu-search task was inspired by optimal stopping problems, like the well-known "secretary problem"[76]. These tasks encourage subjects to balance the goal of choosing the best offer possible against the time costs of evaluating each offer. One reasonable strategy in optimal stopping problems is to compare each offer against some fixed threshold, then choose the first offer that exceeds this threshold. Because offers were uniformly distributed in value and presented at random, there was an identical probability that each offer would exceed a fixed threshold (1 minus the threshold). The distribution for a stopping process with a fixed probability of stopping at discrete steps is a geometric distribution. The maximum likelihood fit of a geometric distribution to the subjects' behavior is illustrated in Fig. 1e (model fit via the expectation-maximization algorithm). The geometric distribution has a single free parameter, which can be expressed equivalently as 1 minus the probability of stopping (here, as the threshold), or its inverse: the average duration of the stopping process (here, the number of offers evaluated per trial). Previously, we calculated the optimal, reward-rate maximizing threshold in this task[11]. Here, we used these mathematical insights to illustrate the distribution over the number of offer evaluations that we would expect from this optimal threshold. Although the geometric distribution was a good fit to the data and the subjects' curve was close to optimal, this task does differ from the classic "secretary problem" because the subjects could return to, reevaluate, and choose a previously-rejected offer at any time, which they often did (Fig. 1f). In short, while the subjects often viewed something close to the optimal number of unique options per certain models of task performance, they were suboptimal in returning to reevaluate previous options and they also appeared to at least occasionally compare options with each other rather than always against a fixed threshold (Fig. 1i, j).

To determine how well choice behavior was predicted by offer values, we fit a four-parameter logistic model. The probability of accepting an offer was modeled as a function of the value of that offer according the following equation:

$$p(\text{chosen}) = \frac{\text{scale}}{1 + \exp(-\text{slope}(\text{value} - \text{intercept}))} + \text{offset} \quad (1)$$

Where "value" was the objective value of the offer on the screen, the intercept captured the value where choices were evenly split between accept and reject, the slope reflected the noise around this intercept (i.e., decision temperature), and the scale and offset reflected the tendency to reject or accept offers by chance, respectively. The model was fit with the default settings of the Matlab fit.m function (minimizing nonlinear least squares, trust region algorithm) and the adjusted $R^2$ was taken as an index of the quality of model fit.

## Neuronal tuning

To identify which neurons were tuned for value, we used an approach that makes no assumptions about the shape of neuronal value tuning. Specifically, we calculated mutual information between the firing rate and the values for each individual neuron:

$$I(X;Y) = \sum_{x \in \mathcal{X}, y \in \mathcal{Y}} p(x,y) \log\left(\frac{p(x,y)}{p(x)p(y)}\right) \quad (2)$$

Where $X$ are the firing rates and $Y$ are the values. Firing rates and values were both divided into quantile bins, where the number of bins (3 each) was chosen to minimize the number of empty cells (unobserved combinations of firing rate and value) while still allowing for tuned neurons with U-shaped value tuning functions. We wanted to avoid empty cells because these can inflate mutual information estimates: it is not clear whether the probability of an empty combination of variables is truly zero or if it simply was not observed within the finite sample of data. We then compared the mutual information in the vmPFC data against a distribution in which the labels had been shuffled (n = 1000 shuffles).

To determine the shape of tuning functions within the set of tuned neurons, we used Mandel's fitting test[77,78]. This method asks if a mean firing rate model that permits curvature (i.e., includes a quadratic term) is a significantly better fit than a model that does not (i.e., a model that includes only linear terms) via calculating the following F-statistic:

$$F = \frac{(n-2)\text{RMSE}^2_{\text{linear}} - (n-3)\text{RMSE}^2_{\text{quadratic}}}{\text{RMSE}^2_{\text{quadratic}}} \quad (3)$$

Where RMSE stands for root mean squared error of the linear and quadratic regression models, respectively, and n is the number of value bins. Because this analysis assumes normally distributed data, offers were quantile-binned into 25 distinct values and the models were fit to average firing rates in each bin.

To determine the shape of the curvilinear tuning across individual neurons, we fit piecewise linear models to each neuron. This allowed us to identify neurons in which curvature was caused by floor effects (downward rectified, with a 0 slope for low values), ceiling effects (upwards rectified, with a 0 slope for high values), or peaking at some intermediate value (opposing slopes for high and low values). The piecewise regression model was fit with the breakpoint as a free parameter, constrained to fall between the 20% lowest and the 20% highest values of the full value range (bins 5–21) to ensure that a sufficient number of datapoints were included to estimate the slope of each half of the function. These results are included in Supplementary Fig. S5. We repeated Mandel's fitting test to compare piecewise linear and quadratic fits.

## Pseudopopulations

Neurons in this study were recorded largely separately, so to gain insight into the representational geometry of value coding at the population level, we built pseudopopulations from non-simultaneously recorded neurons[33–37]. The pseudopopulation approach does not permit a reconstruction of the covariance structure between simultaneously recorded neurons, but it can still be useful for generating first-order insights into how population activity changes across various conditions.

Offers were first quantile-binned into 25 distinct values, a level at which each value bin spanned a ±2% change in reward. Within each bin, firing rates from separately recorded neurons were randomly drawn with replacement to create a pseudotrial firing rate vector, with each entry corresponding to the activity of one neuron when an offer within that quantile bin was on the screen. Pseudotrial vectors were then stacked into a trials-by-neurons pseudopopulation matrix. Twenty-eight pseudotrials were drawn from each cell for each condition because at least 95% of pairs of cells and conditions had at least this number of observations. One neuron was excluded because it did not spike within the selected epoch. All pseudopopulation results are reported for a single, randomly-seeded pseudopopulation, but were later confirmed with a range of different random seeds. For analyzing only accepted and only rejected offers, accept-only and reject-only offers were quantile-binned into 10 distinct values. The number of bins was chosen to account for the smaller number of observations per bin when the data was split according to choice (95% of pairs of cells and

conditions contained at least 28 observations, as for the analysis of all choices together). In order to determine if value encoding depended on the value of early offers within the trial (Supplementary Fig. S13), we made two separate pseudopopulations by splitting trials into 2 groups, based on whether the first offer revealed was below or above 0.5. Two pseudopopulations were then constructed as normal within each of these subsets.

## Linearized control populations

In order to determine if the apparent curvature in vmPFC populations was just an artifact of some data processing step(s), we generated 1000 bootstrapped estimates of the distributions of certain statistical measures under the null hypothesis that the neural representation of value was linear. We did this via linearizing pseudopopulation responses (after[20]). A line was fit to the neural population data, then interpolation was used to identify an evenly spaced set of neural states along that line. To generate realistic trial-to-trial noise, spiking observations were then drawn from Poisson distributions parameterized by the neural states.

## Decoding models

To decode offer values from the neural population data, we fit general linear models of the form:

$$value = X\beta \quad (4)$$

Where $X$ is the matrix of population data (n-trials by k-neurons), augmented by a column of ones to serve as an intercept, $\beta$ is a vector of weights for each neuron (or intercept), and value is one of 25 value bins, with the range rescaled between 0 and 1. Decoding models were trained via maximum likelihood using standard Matlab libraries (glmfit). Decoding was performed via projecting the vmPFC data onto the optimized $\beta$ vector. In order to maximize the accuracy of the models' predictions–and thus the accuracy of our estimate of their residual errors–decoding models were trained and tested on complete data unless otherwise specified.

## Simulated curved and linear populations

In order to test the hypothesis that curvature could introduce systematic biases in choice decoding accuracy (Fig. 5a), we simulated data from populations of neurons that either had curvature or did not have curvature. For the linear population (no curvature), neuronal responses were modeled as a linear function of value:

$$\lambda = \beta_0 + \beta_1(value), \quad \text{spike count} \sim \text{Poisson}(\lambda) \quad (5)$$

To introduce curvature into the neurons in the second population, a quadratic term was added:

$$\lambda = \beta_0 + \beta_1(value) + \beta_2\left(value^2\right) \quad (6)$$

Simulated populations contained 100 cells with 50 observations drawn at each value level (25 value bins, 1250 total observations). The cells' slope and curvature parameters were drawn uniformly at random from the approximate range of values in the vmPFC data ($\beta_1 \in [-0.2, 0.2]$, $\beta_2 \in [-0.02, -0.02]$). Note that the baselines ($\beta_0$) were slightly elevated ($\beta_0 \in [10, 30]$) in order to control for the curvature that can be produced by Poisson spiking when close to the noise floor[20].

## Decoy effects in neural activity

To determine if curvature in the vmPFC population was necessary and sufficient to cause the decoy effect on choice accuracy, we decoded option values from 50 randomly drawn pseudopopulations and their linearized counterparts, and then from 50 simulated populations that were either linear or curved by design. From each population we randomly drew 200 sets of neural population responses to between 3

and 7 unique options. Sets (or "trials") always included two good offers (e.g., value bins 24 and 25 at the high end of the range (Fig. 5) or value bins 12 and 14 [Supplementary Fig. S7]) with the remaining offers chosen at random from the lower portion of the value range. To find the upper bound on choice decoding accuracy for each choice set, we used the decoding model described above. The "decoding accuracy" was defined as the percent of "trials" where the model correctly identified the best option as being more valuable than the 2nd best option, i.e., the likelihood that the decoded highest-value option matched the objectively best option. Decoding accuracy was calculated within each population, for each choice set. The measured "decoy effect" for each population was then the effect of the worst option's value on decoding accuracy (i.e., the slope of decoding accuracy by the minimum value in the set).

## Decoy effects in behavior

To determine whether the subjects had the same irrational decoy effects we found in the neural and simulated data, we asked if the value of irrelevant decoy options also altered choice. We first isolated the trials in which subjects viewed at least 3 options (16,933 total trials; subject J: 9196; subject T: 7737; 26.5% of trials) because these were the only trials in which the subject had seen a non-overlapping best option, second-best option, and a worst, decoy option.

Because subjects had little difficulty choosing the best option when the best option was very high and/or the difference between best and second-best option was large (e.g. they chose the best option 99% of the time when its reward was at least 0.2 units greater than the second-best, but only 70% of the time within this range; see Fig. 1 and Supplementary Fig. S12), we additionally repeated our analyses on the trials where the difference between the best and second-best option were within 0.2 (Supplementary Fig. S10; 8157 total trials; subject J: 4037; subject T: 4120; 5,803 trials which had at least 3 options revealed, subject J: 2937; subject T: 2866).

To determine whether the value of the irrelevant decoy option affected the probability of choosing the best option, we fit a GLM that included the best option's value, the difference between the best and second-best options, and the main effects of the decoy option's value, together with its pairwise interactions with the two other factors:

$$\log\left(\frac{p(\text{chose best})}{1 - p(\text{chose best})}\right) = \beta_0 + \beta_1 V_{\text{best}} + \beta_2\left(V_{\text{best}} - V_{\text{2nd}}\right) + \beta_3 V_{\text{decoy}}$$
$$+ \beta_4\left(V_{\text{best}} \times V_{\text{decoy}}\right) + \beta_5\left(\left(V_{\text{best}} - V_{\text{2nd}}\right) \times V_{\text{decoy}}\right)$$
$$(7)$$

Where $V_{\text{best}}$ is the value of the best option, $V_{\text{2nd}}$ is the value of the second-best option, and $V_{\text{decoy}}$ is the value of the decoy option. To ensure that potential errors in choosing the best option are made among the good options, and not caused by choosing the decoy, we excluded trials in which subjects chose the decoy option (n = 472, 3%). The model was fit with L2 regularization (lambda = 1) to ensure the stability of parameter estimates. Analogous analyses were also performed for probability of making a return within a trial, i.e. making a repeated reveal of a previously seen option.

Several additional factors were taken into account to rule out the possibility that the decoy value effects were an artifact of some aspect of our task design. We tested the effects of the set size (3–7), the order in which the decoy option was presented (first, second, etc.), and the recency of the decoy option with respect to the final choice (one-back, two-back etc.) by adding each of these factors, and its pairwise interactions with the decoy effect, to the GLM described above (Supplementary Fig. S11).

## Reporting summary

Further information on research design is available in the Nature Portfolio Reporting Summary linked to this article.

## Data availability

All the data used in this study has been deposited on OSF (https://doi.org/10.17605/OSF.IO/8MKRD[79]). The data was previously used in[11].

## Code availability

All the code used for data analysis in this study is available on OSF (https://doi.org/10.17605/OSF.IO/8MKRD[79]).

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

## Acknowledgements

This work was supported by a NSERC Discovery Grant (RGPIN-2020-05577), the Research Corporation for Science Advancement & Frederick Gardner Cottrell Foundation (Project #29087), the Canada Research Chair Dynamics of Cognition (FD507106), a Jacobs Foundation Research Fellowship, and a Young Investigator Award from the Brain and Behavior Research Foundation (#27298) to R.B.E., a postdoctoral fellowship from the Institute for Data Valorisation (IVADO) to K.J., and a National Institutes of Health R01 (DA037229) to B.Y.H.

## Author contributions

Conceptualization, R.B.E., B.Y.H., K.J., P.S.M., Formal analysis, K.J., R.B.E., B.J.S., Investigation, P.S.M., Methodology, R.B.E., Visualization, K.J., R.B.E., Writing – original draft, R.B.E., K.J., B.Y.H., Writing – review and editing, K.J., R.B.E., Funding acquisition, R.B.E., B.Y.H., K.J., Supervision, R.B.E., B.Y.H.

## Competing interests

The authors declare no competing interests.
