## [Peer Review File · Nature Communications]

Irrational choices via a curvilinear representational geometry for valueReviewers' Comments:

Reviewer #1:

Remarks to the Author:

Jurewicz et al propose an idea that links curvilinear neural geometry to irrational decisions in value-based decision making. They recorded from vmPFC while monkeys performed a foraging task where they sequentially sampled the value of options and selected one of them. They found neurons in vmPFC encoded the offered value non-linearly. Some neurons showed quadratic tuning and, at the population level, neural population encoded the values on a curvilinear manifold. Assuming linear decoder, the authors argue that this form of encoding leads to suboptimal decisions. Two proximal options can be linearly decoded from this curved manifold, but with the third decoy option, a linear decoder that encompasses all three options poorly discriminates the two proximal options. The authors confirmed that monkeys' behavior was consistent with this decoy effect.

The paper proposes an interesting idea. I have a couple of questions/concerns regarding the key conclusions of the paper.

1) I wasn't clear why the authors used simulated neural data to test their main hypothesis that curvilinear manifolds result in decoy effects. Fig. 5B-C is from simulated data. In the main text, the authors wrote "Critically, we also observed a decoy effect in decoding accuracy when decoding from the real vmPFC population, likely because of the curvature in the population manifold", so they actually tested the same effect with the real data? I think more details should be explained in this part, and the results should be presented in a main figure. According to Fig. 4e, the outcome looks promising, but it might depend on the details of the analysis procedure and the limited amount of data. More information is needed to support the authors' main claim.

2) It is not clear how decoy effects arise in the kind of sequential decision-making task the authors used. Typical multi-alternative tasks offer multiple options simultaneously. In the authors' task, the animals see each option one at a time and decide whether to accept it or not. After seeing 3 options, do monkeys go back to a previously revealed option and choose the best option they found? Then, it looks more or less like a multi-alternative task, so I understand that decoy effects can affect performance. But according to Fig. 1E, such cases seem limited. In the trials where the monkeys revealed only 3 options, was the probability of accepting/rejecting the 3rd option affected by the value of the decoy? Then, the monkeys were deciding whether to accept the 3rd option or not, so they were not really comparing the best vs. 2nd best options, so how does the decoy effect arise? The authors found clear effects of the recency and order of the decoy in Table S3, which confuses me even more about how to interpret their data. Although the authors showed these confounds do not account for the main effect (Table S5), the paper should better clarify how decoy effects arise in their task setting. Perhaps plotting the effects (Fig. 5D-F) in conjunction with the order of the decoy would be helpful.

3) Does vmPFC activity itself depend on the sequential ordering of options or previous options? I think the authors' decoding hypothesis assumes that vmPFC stably encodes value regardless of these contexts and that only the decoder changes depending on contexts. But I have not found empirical evidence for this part.

Reviewer #2:

Remarks to the Author:

In this manuscript, Jurewicz et al. argue that neuronal activity in the vmPFC represents value in a curved geometry. The authors use single-neuron and population-level analyses, as well as simulations, to argue that a curved, rather than linear, representation more closely describes the neural patterns observed in the vmPFC. Furthermore, they describe how readout of a curved representation would result in compression of information at the extremes of value when a larger range of values are

present, which could explain observed difficulties in discriminating between similar high values when a lower option is also available. Overall, the analysis approach rigorous and thoughtful. This is also a uniquely designed task that allows the authors to test neural responses to a range of values presented. However, I have one major concern about a potential confound in interpretation, as well as some minor comments.

Major:

The main claim of the paper is that value is encoded by vmPFC on a curved, rather than linear manifold, and that at least one irregularity in decision-making – effects of irrelevant third options – can be attributed to this curved shape. The premise of the latter point is that a linear decoder might functionally shift around the curved manifold to decode different subsets of values appearing at the same time, and when this line is positioned to capture widely spaced values, there is compression in the readout at the end of the curve, resulting in less reliable choices. This stands in contrast to other proposals in the literature that suggest that the neural responses themselves – not the readout – are altered by the introduction of low-value alternatives, either by range adaptation or divisive normalization. If vmPFC responses were altered by the ‘menu’ of values available, this could also create the illusion of a curved manifold when responses are averaged across trials. Consider neuron responses to two high-value items that undergo a form of range-adaptation based on available options. When a neuron such as this responds to the two high-value items in the presence of a middle value, there will be relatively little adaptation, compared to when the same neuron responds to the same two items in the presence of a low value, in which case they range-adapt and the top two values are coded in a compressed manner. There would be similar effects for range-adapting neurons responding to low-value items. However, when a range-adapting neuron responds to two middle value items, the third option can differ from these middle values by only so much, since there is a finite range of values in the task, so the responses to these middle values will, on average, be less compressed. That is, depending on the alternatives available on each trial, high and low values will sometimes be compressed (when there is a distant 3rd option as part of the set), while middle values won’t include these conditions. Averaged across trials, this could produce the illusion of a curved manifold, when really it’s just trial-by-trial range adaptation of individual neurons. The fundamental difference in these scenarios is that one view proposes a static value response, with 3rd option effects arising as a result of read-out mechanisms; the other proposes an adaptive value response with static read-out. Given the data set the authors have here, they are poised to investigate this possibility and should do so, as it could alter the interpretation of their data.

Minor

For non-linear responses, have the authors considered/tried other functions besides quadratics, for example exponentials?

For the PCA analysis, can you somehow depict neuron loadings to confirm that PC1 and 2 are capturing population variance, rather than just loading on one or two neurons?

Typo: line 553 is not finished

Reviewer #3:

Remarks to the Author:

Jurewicz and colleagues studied the neural coding of reward value in the ventromedial prefrontal cortex (vmPFC) using previously collected data in monkeys performing a decision-making task where they accept or reject sequential visual stimuli associated with different magnitudes of fluid reward. They examined the neural response to these visual stimuli, and found that most of individual neurons

that encoded reward value did so in a linear fashion, with some individual neurons exhibiting nonlinear encoding. They used dimension reduction (PCA) to examine the geometry of the population representation of reward value, and found that a low-dimensional representation was curvilinear. The authors use decoding analyses to argue that the curvilinear geometry predicts a pattern of behavioral effects where monkeys more accurately accept the stimulus associated with largest reward when the value of an irrelevant stimulus (of lower value) is increased. The manuscript is clear and well-written. It offers a refreshing perspective on value coding, which has potential implications for understanding how different types of "irrational" behaviors are generated.

While 38% (46/122) neurons were modulated by reward value, most of these were modulated in a linear fashion. 16% required a quadratic term to fit the data, but how many of these were really non-monotonic? The authors show one clearly non-monotonic example (Fig. 2, cell 26), while the two other non-monotonic example neurons look like half-wave rectification that may be due to floor effects of very low firing rates for lower reward values. The average across all neurons itself resembles a half-wave rectifier (Fig. 2E), and I wonder whether the curved population representation was mostly due to a sufficient number of neurons with floor effects due to low firing rates, or rather due to neurons that were "tuned" for certain reward sizes.

The authors indicate that decoding accuracy of the real vmPFC population exhibited a decoy effect (L. 363). This would be really nice to see in Fig. 5, as it is the most directly comparable analysis relating neural activity to the behavioral decoy effect. Currently that figure shows simulated data (Fig. 5 B-C), although it would be more useful to see the decoding results for real data (with the linearized representation as well). The simulations could be made supplementary.

Another question about Figure 5 is why the worst offer value (E-F) stops at 0.5. Is it simply that there is not much data above this point because the animals always accept the initial stimulus? Also, in these panels the difficulty of the best/2nd best discrimination is fixed by restricting to a 20% difference, but it appears that the data are collapsed over the range of best offer values. However, this seems like a relevant parameter for behavior. I.e., does the decoy effect vary depending on the best offer value (test by including this or average of best and 2nd best value in GLM)? The model in Fig. 5A suggests not, a decoy close in value to best and 2nd best offers that have high values should have the same improved discriminability as a decoy close in value to best and 2nd best offers with low values.

Minor

Although monotonicity is evoked in the Results when referring to the average response across all neurons (L. 152), the authors state that this is a linear relationship in the Discussion (L. 432), contrasting this with the nonlinearity observed in some individual neurons. Linearity was not actually tested for the average, which doesn't actually look linear, so this sentence should be modified to reflect better the figure.

Do "offer size" in Fig. 1 and "offer value" in Fig. 5 index the same measure, the latter being divided by 100? If so, maybe make the language and scale consistent.

In some of the tables the number of samples is listed. Would be nice if this was done for all the model summaries. Also, it is unclear where $n=5803$ in Table S5 comes from?

Reviewer #1:

Jurewicz et al propose an idea that links curvilinear neural geometry to irrational decisions in value-based decision making. They recorded from vmPFC while monkeys performed a foraging task where they sequentially sampled the value of options and selected one of them. They found neurons in vmPFC encoded the offered value non-linearly. Some neurons showed quadratic tuning and, at the population level, neural population encoded the values on a curvilinear manifold. Assuming linear decoder, the authors argue that this form of encoding leads to suboptimal decisions. Two proximal options can be linearly decoded from this curved manifold, but with the third decoy option, a linear decoder that encompasses all three options poorly discriminates the two proximal options. The authors confirmed that monkeys' behavior was consistent with this decoy effect.

The paper proposes an interesting idea. I have a couple of questions/concerns regarding the key conclusions of the paper.

We appreciate the reviewer's detailed summary and comment on the interesting ideas in our study. We believe that the revised manuscript addresses the important questions and concerns that were raised.

1) I wasn't clear why the authors used simulated neural data to test their main hypothesis that curvilinear manifolds result in decoy effects. Fig. 5B-C is from simulated data. In the main text, the authors wrote "Critically, we also observed a decoy effect in decoding accuracy when decoding from the real vmPFC population, likely because of the curvature in the population manifold", so they actually tested the same effect with the real data? I think more details should be explained in this part, and the results should be presented in a main figure. According to Fig. 4e, the outcome looks promising, but it might depend on the details of the analysis procedure and the limited amount of data. More information is needed to support the authors' main claim.

We agree with the reviewer's point that it would be helpful to include predictions made from real data as well as simulated neural populations. Our original motivation for decoding from simulated neural populations was to test if curvature of the manifold was necessary to warp decoding. Simulation allowed us to generate populations that had curvature and populations that verifiably had none (as opposed to our linearization procedure, in which some curvature remained just due to Poisson spiking: this is evident in **Figures 3-5**). We had also confirmed that the curvature in the real vmPFC population was sufficient to produce decoy effects in decoding to a greater extent than in linearized populations, but did not report these results in the paper. These results are now reported in **Figure 5**, alongside the simulated results. **Supplementary Figure 5** repeats both methods using different ranges of values and the **Methods** contains additional detail of these analyses as follows (p. 27, lines 882-896):

"To determine if curvature in the vmPFC population was necessary and sufficient to cause the decoy effect on choice accuracy, we decoded option values from 50 randomly drawn pseudopopulations and their linearized counterparts, and then from 50 simulated populations that were either linear or curved by design. From each population we randomly drew 200

sets of neural population responses to between 3 and 7 unique options. Sets (or “trials”) always included two good offers (e.g. value bins 24 and 25 at the high end of the range [Figure 5] or value bins 12 and 14 [Supplementary Figure S5]) with the remaining offers chosen at random from the lower portion of the value range. To find the upper bound on choice decoding accuracy for each choice set, we used the decoding model described above. The “decoding accuracy” was defined as the percent of “trials” where the model correctly identified the best option as being more valuable than the 2nd best option, i.e. the likelihood that the decoded highest-value-option matched the objectively best option. Decoding accuracy was calculated within each population, for each choice set. The measured “decoy effect” for each population was then the effect of the worst option’s value on decoding accuracy (i.e. the slope of decoding accuracy by the minimum value in the set).”

These results are now reported in the main text as follows (p. 16, lines 442-458):

“We found that curvature in the value-encoding manifold was sufficient to produce strong decoy effects in decoding accuracy (Figure 5B-C; curved population, mean slope of the decoy effect = 0.17, range = [0.02, 0.40], $p < 0.0001$, one-sample t-test from 0). Linearized populations also produced a weaker but positive decoy effect (mean slope of the decoy effect = 0.05, range = [-0.13, 0.18], $p < 0.0001$), suggesting that the linearized population still maintained a level of curvature capable of affecting the readout, likely due to the Poisson spiking close to the noise floor (Okazawa et al. 2021). This result was not an artifact of considering some especially wide range of values: curvature in value-encoding manifold was sufficient to produce strong effects on decoding accuracy when we only considered a narrower range of values (Supplementary Figure S5). To test the role of curvature in producing the decoy effect we additionally simulated 50 curved and linear populations with slightly elevated baseline firing rates to remove them from the noise floor (Supplementary Figures S6, S7). Here, we found that curvature in the manifold was necessary to produce decoy effects in decoding accuracy, as the decoy effect was absent in simulated linear populations (curved population, mean slope of the decoy effect = 0.191, range = [0.009, 0.508], $p < 0.0001$; linear population, mean slope of the decoy effect = 0.005, range = [-0.079, 0.121], $p = 0.436$).”

Figure 5. Curvilinear manifolds suggest a specific pattern of irrational decisions. A) A cartoon illustrating how the ability to discriminate good offers could depend on the value of a worst, “decoy” offer in a curved manifold. The linear decoder represents an upper bound on the accuracy of a downstream neuron’s readout from vmPFC. The inset graph shows how the probability of choosing the best option changes as a function of decoy offer value in an example vmPFC pseudopopulation; the slope here is the “decoy effect”: the effect of the worst offer’s value on the accuracy of the choices (marked in [B] with an open arrow). **B)** Distributions of decoy effect slopes from vmPFC pseudopopulations (black) and their linearized version (purple). Filled arrows = means, open arrow = the example effect illustrated in (A). **C)** Same as (B) but the distributions of decoy effects were obtained from simulated populations that had either a curved (black) or linear (purple) value manifold.

Figure S5. Decoding from a constrained range of values in vmPFC (real and linearized), related to Figure 5. A) A cartoon illustrating how the ability to discriminate good offers could depend on how bad the worst offer is in a curved manifold, when the best values in the set are around the middle of the full range of the manifold and the decoding range is thus constrained (black horizontal braces). **B)** Distributions of decoy effects from vmPFC pseudopopulations (black) and their linearized version (purple) value manifold constrained to the lower 56% of values (value bins 1-14 from the full range of 25, see **Methods**, vmPFC population, mean decoy effect slope = 0.326, range = [-0.003, 0.588], $p < 0.0001$, linearized population, mean decoy effect slope = 0.111, range = [-0.103, 0.377], $p < 0.0001$). Filled arrows indicate the means of the distributions.

2) It is not clear how decoy effects arise in the kind of sequential decision-making task the authors used. Typical multi-alternative tasks offer multiple options simultaneously. In the authors' task, the animals see each option one at a time and decide whether to accept it or not. After seeing 3 options, do monkeys go back to a previously revealed option and choose the best option they found? Then, it looks more or less like a multi-alternative task, so I understand that decoy effects can affect performance. But according to Fig. 1E, such cases seem limited. In the trials where the monkeys revealed only 3 options, was the probability of accepting/rejecting the 3rd option affected by the value of the decoy? Then, the monkeys were deciding whether to accept the 3rd option or not, so they were not really comparing the best vs. 2nd best options, so how does the decoy effect arise? The authors found clear effects of the recency and order of the decoy in Table S3, which confuses me even more about how to interpret their data. Although the authors showed these confounds do not account for the main effect (Table S5), the paper should better clarify how decoy effects arise in their task setting. Perhaps plotting the effects (Fig. 5D-F) in conjunction with the order of the decoy would be helpful.

We agree with the reviewer that a more detailed treatment of the nature of decisions in this task would enable a deeper understanding of the behavior and clarify the interpretation of our results. The paper now reports that the animals used a combination of strategies in which they evaluated the value of each option as it was encountered, but then also compared later values against the set of options they had seen so far, regularly returning to previously evaluated options. (We show through a variety of analyses (**Figure 1**; **Figure 5**; **Supplemental Figures 8-10**) that these returns were related to uncertainty about which option to choose and that they improved performance.) Critically, these returns also represent a second, largely independent, and more sensitive measure of the same decoy effect reported in the original manuscript: we found that subjects were also more likely to vacillate between good options when the decoy was low-value, compared to when it was high.

To illustrate this additional insight, we restructured the report of the decoy effects to include both the probability of choosing the best option and the probability of making a return as a

function of the worst option in the set (**new Figure 5, Supplementary Figures S8 and S10, Supplementary Tables S2 and S3**).

To the reviewer's second point about needing to provide additional detail about any possible order and recency effects, we now illustrate the decoy effects within particular values of the variables in question (decoy order, recency, and additionally the number of offers viewed in the trial, **Supplementary Figure S9**). Overall, this figure illustrates that the decoy effect is quite robust, but indeed strongest in those trials when we would expect it to be strongest: when good values are most difficult to discern, when the decoy is presented either early on in the sequence or soon before the time of the choice. Adding these confounding variables to our main GLM analyses confirmed that in each case, the decoy effect was not an artifact of these potential confounds.

Details of the returning behavior and strategies observed in the task are described in the Results as follows (p. 5-6, lines 127-163):

“Importantly, the design of this task allowed subjects to return to reevaluate a previously revealed offer at any point. Subjects returned at least once in ~10% of trials (**Figure 1F**; subject J: 11.81+/- 0.73%; subject T: 7.04 +/- 0.55% SEM, both $p < 0.0001$, one-sample t-test from 0). Trials in which returns occurred (return trials) tended to have a lower expected value than strictly sequential trials, suggesting that returns happened when no offer was clearly above threshold or best (**Figure 1G**; return trials, subject J: 0.48+/- 0.01; subject T: 0.37 +/- 0.01; trials without returns, subject J: 0.60 +/- 0.01; subject T: 0.58 +/- 0.01, both $p < 0.0001$, two-sample t-test). However, the returns were not indicative of disengagement or accidental repeated selection because they were more likely to return to high-value than low-value offers (**Figure 1H**; average value of the offers subjects returned to, subject J: 0.44 +/- 0.01; subject T: 0.33 +/- 0.01; average value of the remaining offers in the return trials, subject J: 0.25 +/- 0.01; subject T: 0.28 +/- 0.01, both $p < 0.0001$, two-sample t-test). Further, considering only the trials in which subjects revealed multiple unique offers (i.e. at least three), we found that they tended to make one or two returns per trial (**Figure 1I**) and that subjects most often returned to an offer to accept it (though the probability of return also increased towards the end of a trial; **Figure 1J**). Taken together, the pattern of the subjects' returns suggested a kind of "dithering" in the face of uncertainty: a process of doubling back to reconsider one or two good options prior to committing to one of them.

Because the subjects could return to a previously viewed offer, their strategy in this task was best described as a mixture of 2 kinds of decision-making processes: a compare-to-threshold process (where each offer was sequentially evaluated against a threshold and accepted immediately if it was clearly above threshold) and a value-comparison process (where newly encountered offers were compared against previously-viewed and still-available offers). In a compare-to-threshold process, the decision is most difficult if values are not clearly above or below the threshold, in particular, if the best option in forward sweep through the offers is still relatively low. In a value-comparison process, the decision is most difficult when good options are close in value and thus harder to discriminate. We found that the subjects were most likely to make errors (i.e. not choose the best offer) when either form of difficulty was high (**Figure 1K**; best offer value effect, subject J: 2.17, range = [1.17, 3.23], subject T: 2.65, range = [0, 3.81], both $p < 0.0001$, logistic regression; best-2nd best offer value effect, subject J: 2.11, range = [1.12, 3.09], subject T: 2.53, range = [0, 3.58], both $p < 0.0001$). Both subjects were also most likely to return to previously viewed offers when either

form of difficulty was high (**Figure 1L**; best value, subject J: -1.65, range = [-3.28 - 0.04], subject T: -1.94, range = [-3.25, -0.65], both $p < 0.0001$; best-2nd best, subject J: -1.33, range = [-2.60, 0.95], subject T: -1.24, range = [-2.46, -0.12], both $p < 0.0001$). In sum, both subjects evaluated close to the optimal number of unique offers in each trial, but they did so via a mixed strategy that compared offers both to a fixed threshold and to previous offers encountered within the trial.”

Figure 1. Task design and behavior. **D)** Probability of choosing an offer as a function of offer value. Gray line = logistic fit. Each dot represents an offer value. **E)** Distribution of the number of offers viewed per trial. Cyan line = prediction from an optimal compare-to-threshold model. Black line = best fit compare-to-threshold model (maximum likelihood). **F)** Probability of returning to a previously seen offer. **G)** Expected value of trials which did (“ret.” = return) or did not (“seq.” = sequential) contain a return to a previous offer. **H)** Average value of offers which were returned to (“ret.” = return) versus those which were seen only once (“seq.” = sequential). **I)** Probability of a given number of returns in a trial, in trials where multiple unique offers were revealed (>3). **J)** Probability of the return in multiple offer trials as a function of the viewing time at which an offer was finally accepted (at position 0). **K)** Probability of choosing the best option as a function of value of the best offer revealed within a trial (left) or as a function of the difference in value between the best and the 2nd best offer (right), multiple offer trials. **L)** The same as (K) but for returning to a previously seen option. Lines = least squares fit. Error bars indicate \pm standard error of the mean across sessions (SEM).

The probability of return as a function of the worst option in the set (decoy effect in returning behavior) is described in the Results as follows (p. 17, lines 479-495):

“Errors of reward maximization are only one of the consequences of uncertainty about the best option in the task. As described in **Figure 1**, the monkeys were also more likely to return to previous options on trials where the best decision was uncertain. Therefore, we reasoned that any effect of decoys on the discriminability of good options or of good options from threshold might also result in an increase in returns. As predicted, subjects were less likely to dither between options (i.e. they made fewer returns) when the value of the worst option in the set got higher (**Figure 5G-I**). This was true even after taking into account the best offer value and the difference between the best and 2nd best offers (both subjects: mean slope of decoy effect slope = -0.45, range = [-1.24, 0.52], $p < 0.0001$; same GLM as for accuracy, similar results for individual subjects, **Supplementary Table S2**). Again, the decoy effect was most pronounced for lower best offer values and best offer values close to the 2nd best offers (**Figure 5I**; both subjects: decoy by best interaction, mean slope = -0.43, range = [-1.01, 0.48], $p < 0.0001$; decoy by best-2nd best interaction, mean slope = -0.18, range = [-0.39, 0.10], $p < 0.0001$; similar results for individual subjects, **Supplementary Table S2**; **Supplementary Figure S8D-F**). Together, these results converge to suggest that the subjects were more uncertain about which offer to choose when irrelevant decoy offer values were lower, tracking the upper bound on decoding accuracy predicted by the curved manifold.”

Figure 5. Curvilinear manifolds suggest a specific pattern of irrational decisions. D) Probability of choosing the best offer seen as a function of the worst offer seen (decoy offer quantile). Lines = least squares fit. **E)** Distribution of decoy effect slopes across sessions. **F)** Probability of choosing the best option as a function of the decoy value and other variables that affect accuracy: Left) The best value seen. Right) The difference between the best and 2nd best option. **G-I)** Same as (D-F), but for the probability of returning to a previously seen offer. Insets in **G)** show which options tended to be the target of returns. Error bars indicate \pm standard error across sessions (SEM). These may be smaller than the symbols. **** $p < 0.0001$.

Figure S8. Decoy effect within most confusable trials, related to Figure 5. A) Probability of choosing the best offer seen as a function of the worst offer seen within trials, in which the best and 2nd best offer values were < 0.2 from each other (n , subject J = 2,937; n , subject T = 2,866). Line = least squares fit. **B)** Distribution of decoy effects within the probability of choosing the best option in the set across sessions. **C)** Probability of choosing the best option seen within each trial as a function of both the best value and the decoy value in the set (left) and the difference between the best and 2nd best option, and the decoy value (right). **D-F)** Same as (A-C) but for the probability of return. **** $p < 0.0001$.

The relationship between the returns and choosing the best value is summarized in the Results as follows (p. 17-18, lines 512-521):

“Finally, while choice accuracy and the probability of returns were clearly related (**Supplementary Figure S10**), returns did not explain the errors of reward maximization. Decoy effect was still present in choice accuracy within only those trials that did not contain any returns (both subjects: mean slope of decoy effect = 0.05, range = [-0.70, 0.54], $p < 0.0442$; decoy by best interaction, mean slope = 0.20, range = [-0.00, 0.42], $p < 0.0001$; decoy by best - 2nd best interaction, mean slope = 0.23, range = [-0.12, 0.60], $p < 0.0001$ [choosing the best option by best value, best-2nd best value, decoy value, and pairwise interactions with decoy value]). Together, these results suggest that errors and returns both arose when the choices were uncertain, and constitute converging pieces of evidence that low-value decoys warp discriminability of offers.”

Figure S10. Errors and returns depending on the best, best-2nd best and worst offer values (for trials in which at least 3 offers were revealed), related to Figures 1 and 5. A) Distribution of trials in which the best offer was chosen (i.e. the choice was “correct”; gray) or some other offer was chosen (i.e. an “error” was made; red), depending on the value of the best, best-2nd best and worst offer revealed in the trial. **B)** Same as A, but for trials which contained only sequential reveals, (gray) versus those that contained returns (black), depending on the value of the best, best-2nd best and worst offer revealed in the trial. **C)** Relationships between pairs of reward-related variables, split by “correct” (best offer chosen; gray) and “error” (other offer chosen; red) trials. **D)** Same as C, but split by purely sequential (gray) versus return (black) trials. Error bars in each graph indicate \pm standard error of the mean across sessions (SEM). These are sometimes smaller than the symbols.

The robustness of the results to potential confounds (like order, recency, number of options, and trial difficulty) are summarized in the text of the Results as follows (statistical details in Supplementary Figure S9, p. 17-18, lines 497-521:

“Nonetheless, because the task was not designed to test for decoy effects, we considered the possibility that the decoy effects were an artifact of some confounding variable that might be correlated with decoy value. We found decoy value consistently improved choice accuracy and decreased the probability of returns in a way that could not be explained by any differences in set size, decoy order effects, decoy recency effects, distraction by the decoy, or some interaction between accuracy and returns. First, while subjects were more likely to see low-value decoys when they viewed more offers in a trial (sig. correlation, $r = -0.33$, $p < 0.0001$), the decoy value effect was present also after accounting for the number of offers revealed (**Supplementary Figure S9A-C**). Second, decoy value influenced choices and returns across decoy’s position in the trial sequence (**Supplementary Figure S9D-F**) and its recency to the time of the choice (**Supplementary Figure S9G-I**). Third, neither the accuracy or return effects were due to distraction by the worst value. The majority of the returns went to the best or 2nd best option (**Figure 5G**, insets; subject J: 84 \pm 6%, subject T: 72 \pm 2%), with no more attention paid to the worst option than any other options (subject J: 6 \pm 1% vs 9 \pm 1%, $p < 0.0001$, subject T: 13 \pm 1% vs 13 \pm 1%, ns.) and the likelihood

that the subjects chose the worst option was negligible (subject J: 2.09 +/- 0.16% subject T: 3.65 +/- 0.20%). Finally, while choice accuracy and the probability of returns were clearly related (**Supplementary Figure S10**), returns did not explain the errors of reward maximization. Decoy effect was still present in choice accuracy within only those trials that did not contain any returns (both subjects: mean slope of decoy effect = 0.05, range = [-0.70, 0.54], $p < 0.0442$; decoy by best interaction, mean slope = 0.20, range = [-0.00, 0.42], $p < 0.0001$; decoy by best - 2nd best interaction, mean slope = 0.23, range = [-0.12, 0.60], $p < 0.0001$ [choosing the best option by best value, best-2nd best value, decoy value, and pairwise interactions with decoy value]). Together, these results suggest that errors and returns both arose when the choices were uncertain, and constitute converging pieces of evidence that low-value decoys warp discriminability of offers.”

Figure S9. Decoy effect in split for number of offers viewed, decoy order and recency, related to Figure 5. **A)** Percent of trials in which subjects revealed 3, 4 or more (5-7) offers. **B)** Decoy effect for the subset of trials in which subjects revealed 3, 4 or more offers ([GLM that includes number of offers viewed, best value, best-2nd best value, decoy value, and pairwise interactions with decoy value], choice accuracy: mean decoy effect slope = 0.07, range = [-0.22, 0.33], $p < 0.0001$, decoy by number of offers interaction, mean slope = 0.07, range = [-0.79, 1.13], $p < 0.0891$; probability of return: mean decoy effect slope = -0.11, range = [-0.48, 0.24], $p < 0.0001$; decoy by number of offers interaction, mean slope = -0.40, range = [-1.62, 0.61], $p < 0.0001$). **C)** Same as (B), but for the subset of trials in which subjects demonstrated the highest confusion, i.e. trials in which the best and 2nd best offers were within 0.2 value range of each other. **D)** Percent of trials in which decoy was revealed as first, second or third in the sequence. **E)** Decoy effect for the subset of trials with the decoy option presented as first, second, third ([GLM that includes decoy order, best value, best-2nd best value, decoy value, and pairwise interactions with decoy value], choice accuracy: mean decoy effect slope = 0.08, range = [-0.54, 0.52], $p < 0.0001$, decoy value by decoy order interaction, mean slope = 0.09, range = [-1.01, 1.09], $p < 0.0309$; probability of return: mean decoy effect slope = -0.20, range = [-0.73, 0.46], $p < 0.0001$, decoy value by decoy order interaction, mean slope = -0.34, range = [-1.31, 0.52], $p < 0.0001$). **F)** Same as (E), but for the trials in which the best and 2nd best offers

were within 0.2 value range of each other. **G)** Percent of trials in which decoy was revealed as -1, -2, or -3 offer before the choice. **H)** Decoy effect for the subset of trials with the decoy option presented as first, second, third before the last one ([GLM that includes decoy recency, best value, best-2nd best value, decoy value, and pairwise interactions with decoy value], choice accuracy: mean decoy effect slope = 0.13, range = [-0.34, 0.54], $p < 0.0001$, decoy value by decoy recency interaction, mean slope = 0.14, range = [-0.71, 1.04], $p < 0.0005$; probability of return: mean decoy effect slope = -0.26, range = [-0.86, 0.27], $p < 0.0001$, decoy value by decoy recency interaction, mean slope = -0.27, range = [-1.19, 0.88], $p < 0.0001$). **I)** Same as **(H)**, but for the trials in which the best and 2nd best offers were within 0.2 value range of each other. Line = least squares fit. Error bars in each graph indicate +- standard error of the mean across sessions (SEM). These are sometimes smaller than the symbols.

3) Does vmPFC activity itself depend on the sequential ordering of options or previous options? I think the authors' decoding hypothesis assumes that vmPFC stably encodes value regardless of these contexts and that only the decoder changes depending on contexts. But I have not found empirical evidence for this part.

This is an interesting point; we had not thought to ask if the vmPFC manifold differs with context. Although the fact that this is a pseudopopulation study precluded the kinds of single-trial analyses that would be ideal for testing these ideas, we were able to determine whether the 1st offer encountered in the trial set up a "context" that altered the neural representation of subsequent offers (**Supplementary Figure S11**). We did find some evidence for flexibility in the vmPFC manifold; this flexibility was minimal and could not explain the observed decoy effects. Moreover, we found that this kind of context effect would tend to make the decoy effects we report here smaller, not larger.

(Please also refer to the answer to reviewer #2's major point 1)

The context effects analysis is described in the Methods as follows (p. 26, lines 838-842):

"In order to determine if value encoding depended on the value of early offers within the trial (**Supplementary Figure S11**), we made two separate pseudopopulations by splitting trials into 2 groups, based on whether the first offer revealed was below or above 0.5. Two pseudopopulations were then constructed as normal within each of these subsets."

The results are reported in the text as follows (p. 18, lines 523-530):

"We found some evidence for divisive normalization when considering sequential effects in this dataset (**Supplementary Figure S11**): namely, in trials in which the first offer viewed was higher than average, we found a shift in the representation of low-value options such that they would be decoded as slightly higher-value than they were. Because value normalization is linked with the inverse of the decoy effects we report here (i.e. higher decoy values causing worse performance, not better performance), these normalization effects would only cause us to underestimate the magnitude of the effect of the curved value manifold on decision-making."

Figure S11. Average neuronal tuning and the curvature of the manifold within trials that started from low and high value offers, related to Figures 2, 3 and 5. (Left) Average firing rates from all 122 neurons, plotted as a function of value quantile, separately for trials which started from a low (<0.5; blue dots) or a high (≥ 0.5 value; red dots) value offer. Error bars indicate \pm standard error of the mean across neurons (SEM). Lines indicate linear fit. (Right) The projection of the neural population onto the first 2 principal components (PCs), performed separately for trials which started from a low (<0.5; blue dots) or a high (≥ 0.5 value; red dots) value offer. PCA fit was performed for both pseudopopulations together. Lines indicate quadratic fit. The figure legend is shared by both plots.

Reviewer #2:

In this manuscript, Jurewicz et al. argue that neuronal activity in the vmPFC represents value in a curved geometry. The authors use single-neuron and population-level analyses, as well as simulations, to argue that a curved, rather than linear, representation more closely describes the neural patterns observed in the vmPFC. Furthermore, they describe how readout of a curved representation would result in compression of information at the extremes of value when a larger range of values are present, which could explain observed difficulties in discriminating between similar high values when a lower option is also available. Overall, the analysis approach rigorous and thoughtful. This is also a uniquely designed task that allows the authors to test neural responses to a range of values presented. However, I have one major concern about a potential confound in interpretation, as well as some minor comments.

Major:

The main claim of the paper is that value is encoded by vmPFC on a curved, rather than linear manifold, and that at least one irregularity in decision-making – effects of irrelevant third options – can be attributed to this curved shape. The premise of the latter point is that a linear decoder might functionally shift around the curved manifold to decode different subsets of values appearing at the same time, and when this line is positioned to capture widely spaced values, there is compression in the readout at the end of the curve, resulting in less reliable choices. This stands in contrast to other proposals in the literature that suggest that the neural responses themselves – not the readout – are altered by the introduction of low-value alternatives, either by range adaptation or divisive normalization. If vmPFC responses were altered by the ‘menu’ of values available, this could also create the illusion of a curved manifold when responses are averaged across trials. Consider neuron responses to two high-value items that undergo a form of range-adaptation based on

available options. When a neuron such as this responds to the two high-value items in the presence of a middle value, there will be relatively little adaptation, compared to when the same neuron responds to the same two items in the presence of a low value, in which case they range-adapt and the top two values are coded in a compressed manner. There would be similar effects for range-adapting neurons responding to low-value items. However, when a range-adapting neuron responds to two middle value items, the third option can differ from these middle values by only so much, since there is a finite range of values in the task, so the responses to these middle values will, on average, be less compressed. That is, depending on the alternatives available on each trial, high and low values will sometimes be compressed (when there is a distant 3rd option as part of the set), while middle values won't include these conditions. Averaged across trials, this could produce the illusion of a curved manifold, when really it's just trial-by-trial range adaptation of individual neurons. The fundamental difference in these scenarios is that one view proposes a static value response, with 3rd option effects arising as a result of read-out mechanisms; the other proposes an adaptive value response with static read-out. Given the data set the authors have here, they are poised to investigate this possibility and should do so, as it could alter the interpretation of their data.

The reviewer argues that range adaptation could underlie the decoy effects because range adaptation in individual neurons might produce the illusion of a curved manifold when trial averaging is used. We disagree. We have attempted to verify the reviewer's theory by simulating range-adapting neurons and averaging them across trials, but did not find that this was sufficient to cause the illusion of curvature. (Divisive adaptation only scaled where the values lie along the original linear axis and the amount of noise along other principle axes; this noise washed out with larger simulations and did not appear to be sufficient to cause curvature.) Instead, based on previously published work (Okazawa et al. 2021) and our own observations in simulated populations here (e.g. **Figure 5**; **Supplemental Figure S6**), we would argue that low-level properties of spiking statistics are sufficient to produce curvature at the population level. Given that the population will be curved simply because spikes are discrete and have ceiling and floor effects, we do not believe it would be prudent to invoke more complex or speculative explanations for the curvature observed here. That said, the revised manuscript does shore up the argument that the manifold is more curved than we would expect from Poisson spiking alone. For example, we find single neurons with obviously u-shaped tuning functions (**Figure 2**; **Supplemental Figure S3**) and the curvature we observe in the vmPFC population is consistently more pronounced than what we observe after our linearization procedure (which preserves Poisson noise; **Figure 3-5**).

To the reviewer's point that a flexible readout scheme seems like a convoluted assumption to make, we agree and realize that our discussion was not sufficiently clear on this point. Our argument is only that curvature places an upper bound on decoding accuracy, not that readout is flexible. We have revised the relevant passage of the discussion as follows (p. 19, lines 587-601):

"The second premise that our hypothesis may appear to depend on is the idea that a different downstream neuron would be responsible for decoding different sets of options. That is, one could imagine that a downstream region might flexibly alter its decoding strategy in order to maximize information about the currently available set. Given that we know that neural representations are constantly evolving, both spontaneously (Rule et al. 2020; Montijn

et al. 2016) and according to internal states, task contexts, and value ranges (Padoa-Schioppa and Assad 2007; Padoa-Schioppa 2009; Kobayashi et al. 2010; Çukur et al. 2013; Ebitz and Platt 2015; Sohn et al. 2019; Ebitz et al. 2019; Sasaki et al. 2020), this does not seem like an unreasonable idea. However, it is worth clarifying that our argument is not predicated on flexible decoding. Instead, we argue only that the curved manifold creates a bottleneck in information transmission: an upper bound on the decoding accuracy possible in all of the neurons downstream from vmPFC. We struggle to see how it would be possible to compensate for an information bottleneck like this, even allowing nonlinearities and complex network effects, but future work is certainly needed to determine how regions downstream from vmPFC decode from this curved manifold.”

Finally, the reviewer makes the broader point that we should address whether range adaptation / divisive normalization could underlie the results observed in our task, and we agree (see the related point raised by reviewer #1). We have added an additional analysis that clarifies why any effects of divisive normalization would go in the opposite direction of the effects we report here (only making the decoy effects weaker). We have also amended the text to better explain why previous empirical findings and the logic of our task suggest that adaptation and/or normalization are not likely to play a big role in this specific task.

The revisions to our discussion read as follows (p. 20, lines 603-627):

“There is precedent for the idea that single neurons have flexible, context-dependent value representations in the literature on divisive normalization (Louie et al. 2011; Louie et al. 2013; Khaw et al. 2017) and range adaptation (Padoa-Schioppa and Assad 2007; Padoa-Schioppa 2009; Kobayashi et al. 2010). In fact, one might even wonder if the decoy effects here could be due to set-dependent changes in value encoding in single neurons rather than the nonlinear value manifold in vmPFC. However, context or set had only minimal effects on the vmPFC manifold that would have only made the decoy effects weaker, not stronger. Further, neither form of flexible encoding seems like a sufficient alternative explanation for the effects reported here. First, while value normalization can certainly cause interference from low values, it is typically linked with the inverse of these decoy effects: worse performance with higher decoy values, not better performance (Louie et al. 2013). Normalization-like effects are also more typical of task designs when options are presented simultaneously, rather than sequentially, perhaps because divisive normalization is related to competition for visual attention (Gluth et al. 2020). Second, while range adaptation could produce decoy effects with the same sign as those reported here, range adaptation effects typically take place at longer timescales (across blocks or tasks rather than within trials; Padoa-Schioppa and Assad 2007; Padoa-Schioppa 2009; Kobayashi et al. 2010). Significant adaptation would have also compromised the ability to compare sequentially-viewed offers against a stable threshold, but the subjects reliably accepted offers that were above a threshold here. Intriguingly, a curved manifold in vmPFC—a region upstream from the orbitofrontal regions in which range adaptation is maximal (Carmichael and Price 1996)—could actually be part of the as-yet-unknown mechanism(s) for range adaptation. That is, perhaps downstream neurons range-adapt in part because doing so maximizes the information they can decode from nonlinear, curved representational structures. Simultaneous population recordings in vmPFC and orbitofrontal range-adapting neurons might be the key to addressing this question.”

The new analysis is described in the Methods as follows (p. 26, lines 838-842):

“In order to determine if value encoding depended on the value of early offers within the trial (**Supplementary Figure S11**), we made two separate pseudopopulations by splitting trials into 2 groups, based on whether the first offer revealed was below or above 0.5. Two pseudopopulations were then constructed as normal within each of these subsets.”

The results are reported in the text as follows (p. 16, lines 458-463 and p. 18, lines 523-533):

“In short, a curved value manifold predicts a paradoxical behavioral phenomenon where high decoy offers would make it easier to discern which offer in a given set is best. (Note that this is the *inverse* of what a divisive normalization account would predict, where low-value [not high-value] decoys would improve decoding good offer values because lower decoys reduce the magnitude of the divisor component [Louie et al. 2011; Louie et al. 2013].)”

“Again, these effects did not appear to be the simple consequence of divisive normalization. We found some evidence for divisive normalization when considering sequential effects in this dataset (**Supplementary Figure S11**): namely, in trials in which the first offer viewed was higher than average, we found a shift in the representation of low-value options such that they would be decoded as slightly higher-value than they were. Because value normalization is linked with the inverse of the decoy effects we report here (i.e. higher decoy values causing worse performance, not better performance), these normalization effects would only cause us to underestimate the magnitude of the effect of the curved value manifold on decision-making. In sum, what should have been an irrelevant decoy—the worst option in the set—interfered with decision-making the most when it was furthest from the best available options, exactly as we would expect if values were decoded from a curved manifold.”

Figure S11. Average neuronal tuning and the curvature of the manifold within trials that started from low and high value offers, related to Figures 2, 3 and 5. (Left) Average firing rates from all 122 neurons, plotted as a function of value quantile, separately for trials which started from a low (<0.5; blue dots) or a high (>=0.5 value; red dots) value offer. Error bars indicate \pm standard error of the mean across neurons (SEM). Lines indicate linear fit. (Right) The projection of the neural population onto the first 2 principal components (PCs), performed separately for trials which started from a low (<0.5; blue dots) or a high (>=0.5 value; red dots) value offer. PCA fit was performed for both pseudopopulations together. Lines indicate quadratic fit. The figure legend is shared by both plots.

Minor

For non-linear responses, have the authors considered/tried other functions besides quadratics, for example exponentials?

We had not, but now have also considered a piecewise linear fit, which allows us to illustrate the relative fraction of neurons with ceiling / floor saturating tuning curves (vs u-shaped tuning curves; **Supplementary Figure S3**). Because the quadratic is a flexible model that allows us to efficiently estimate the number of nonlinearly-tuned neurons and we did not find a consistent advantage of the piecewise linear fit over the curvilinear fit, we focused on the quadratic fits in the main text.

Figure S3. Piecewise linear regression of firing rates depending on offer value, related to Figure 2. A) Schematic representation of slope orientations in piecewise linear regression. **B)** Each point represents one neuron that was either significantly tuned (filled circles) or untuned (open circles) to the offer value according to the test based on mutual information between firing rates and value labels (see **Methods**). Green points indicate 10 cells with highest absolute PC2 loadings (see **Supplementary Figure S4**). The piecewise linear model fits independent slopes to neural responses to high and low values, with the break point constrained to fall between the 20% lowest and the 20% highest values. Points along the vertical axis only have a slope for high values, points along the horizontal axis only have a slope for low values. There was an overall negative correlation between lower and higher value slopes ($r = -0.28$, $p < 0.002$), supporting the finding of non-linear shapes in single cell responses. Only 19% of the tuned neurons were better described with piecewise linear fit than the curvilinear fit suggesting that floor (downward rectified) or ceiling (upwards rectified) effects were not a better explanation for the majority of the tuned responses.

For the PCA analysis, can you somehow depict neuron loadings to confirm that PC1 and 2 are capturing population variance, rather than just loading on one or two neurons?

Sure. These results are now presented in **Supplementary Figure S4**. In short, there are no abrupt changes in the contributions of different cells to the higher-order (curved) PCs. Eliminating curvature from PC2 requires eliminating approximately 8-10 neurons, but this is not sufficient to remove the curvature from the population manifold (i.e. the other PCs). Thus, no single neuron (or pair of neurons) is responsible for the population curvature, rather, multiple neurons contribute to the population-level effects.

Figure S4. Contribution from single cells to the population manifold, related to Figure 3. A) PC loadings for PC1, PC2 and PC3 across the neurons sorted from highest to lowest. **B)** The projection of the neural population onto the first 2 principal components (PCs) after excluding 2, 4, 8 or 10 of the neurons with highest weights in the PC1. Excluding the neurons with highest PC1 loadings has little impact on the visible curvature of the projection of the manifold. **C)** The projection of the neural population onto PC1 and PC2 or PC1 and PC3 after excluding 2, 4, 8 or 10 of the neurons with highest weights in the PC2 (see also **Supplementary Figure S3** for piecewise linear regression weights for these neurons). Excluding the neurons with highest PC2 loadings removes the curvature from the PC1 vs PC2 projection. However, curvature can still be found in PC3. Shades of gray = value bins from low (light gray) to high (dark gray). Dotted line = best linear fit. Solid line = best quadratic fit.

Typo: line 553 is not finished

We've corrected this typo and appreciate the reviewer's close reading of the manuscript.

The section title now goes as follows (p. 22, line 691):

“Electrophysiological techniques, eye tracking and reward delivery”

Reviewer #3:

Jurewicz and colleagues studied the neural coding of reward value in the ventromedial prefrontal cortex (vmPFC) using previously collected data in monkeys performing a decision-making task where they accept or reject sequential visual stimuli associated with different magnitudes of fluid reward. They examined the neural response to these visual stimuli, and found that most of individual neurons that encoded reward value did so in a linear fashion, with some individual neurons exhibiting nonlinear encoding. They used dimension reduction (PCA) to examine the geometry of the population representation of reward value, and found that a low-dimensional representation was curvilinear. The authors use decoding analyses to argue that the curvilinear geometry predicts a pattern of behavioral effects where monkeys more accurately accept the stimulus associated with largest reward when the value of an irrelevant stimulus (of lower value) is increased. The manuscript is clear and well-written. It offers a refreshing perspective on value coding, which has potential implications for understanding how different types of "irrational" behaviors are generated.

We appreciate the reviewer's kind words about the clarity and novelty of this manuscript.

While 38% (46/122) neurons were modulated by reward value, most of these were modulated in a linear fashion. 16% required a quadratic term to fit the data, but how many of these were really non-monotonic? The authors show one clearly non-monotonic example (Fig. 2, cell 26), while the two other non-monotonic example neurons look like half-wave rectification that may be due to floor effects of very low firing rates for lower reward values. The average across all neurons itself resembles a half-wave rectifier (Fig. 2E), and I wonder whether the curved population representation was mostly due to a sufficient number of neurons with floor effects due to low firing rates, or rather due to neurons that were "tuned" for certain reward sizes.

The reviewer raises an important question that was not sufficiently addressed in the original manuscript: is the curvature just due to floor and ceiling effects in single neurons? Before we describe how we have revised the manuscript to answer this question, we do want to stress that the major phenomena reported here (the population curvature and decoy effects) would emerge irrespective of the source of the curvature in single neuron responses. Previously published work (Okazawa et al. 2021) shows that population-level curvature can be a by-product of the floor and ceiling effects inherent in Poisson-spiking single neurons. The revised manuscript also shows that our procedure for linearizing the population manifold (which preserved Poisson statistics) was not sufficient to entirely remove curvature from the population geometry (**new Figure 5**, see also the responses to reviewer #1 [point 1] and reviewer #2 [major comment]).

To address the reviewer's specific question about the distribution of tuning functions, we fit neuronal value-tuning responses with piecewise linear functions that allow us to plot the different patterns of tuning across the entire population of both tuned and untuned neurons (**Supplementary Figure S3**). Although some neurons do show a downwards or upwards rectification, indicative of floor and ceiling effects respectively, others had clearly u-shaped tuning: with opposing slopes for high and low values. Specifically, among those neurons which had the highest loadings to the curvature captured by PC2, there was a mixture of neurons with inverted u-shaped functions and half-rectified patterns. Thus, while we cannot

rule-out the contribution of single-neuron floor and ceiling effects to the population curvature, contribution from neurons tuned to specific values is at least equally plausible.

Figure S3. Piecewise linear regression of firing rates depending on offer value, related to Figure 2. A) Schematic representation of slope orientations in piecewise linear regression. **B)** Each point represents one neuron that was either significantly tuned (filled circles) or untuned (open circles) to the offer value according to the test based on mutual information between firing rates and value labels (see **Methods**). Green points indicate 10 cells with highest absolute PC2 loadings (see **Supplementary Figure S4**). The piecewise linear model fits independent slopes to neural responses to high and low values, with the break point constrained to fall between the 20% lowest and the 20% highest values. Points along the vertical axis only have a slope for high values, points along the horizontal axis only have a slope for low values. There was an overall negative correlation between lower and higher value slopes ($r = -0.28$, $p < 0.002$), supporting the finding of non-linear shapes in single cell responses. Only 19% of the tuned neurons were better described with piecewise linear fit than the curvilinear fit suggesting that floor (downward rectified) or ceiling (upwards rectified) effects were not a better explanation for the majority of the tuned responses.

The authors indicate that decoding accuracy of the real vmPFC population exhibited a decoy effect (L. 363). This would be really nice to see in Fig. 5, as it is the most directly comparable analysis relating neural activity to the behavioral decoy effect. Currently that figure shows simulated data (Fig. 5 B-C), although it would be more useful to see the decoding results for real data (with the linearized representation as well). The simulations could be made supplementary.

We agree that illustrating prediction from real data would also be helpful (a point also made by reviewer #1), so we now include this in the main figure (**Figure 5**). In short, we find that decoding from the real vmPFC population indeed produced a decoy effect and that this effect was larger than the decoy effect observed in linearized populations. Because our linearization procedure preserved Poisson-spiking statistics, it did not completely remove curvature from the manifold, so we also include the results of simulations in which we were able to generate “neuronal populations” that had no curvature whatsoever in **Figure 5**. Complementary analyses on a constrained set of values are also presented in **Supplementary Figure S5**.

These two, parallel analyses are now described side-by-side in the Methods (p. 27, lines 882-896):

“To determine if curvature in the vmPFC population was necessary and sufficient to cause the decoy effect on choice accuracy, we decoded option values from 50 randomly drawn pseudopopulations and their linearized counterparts, and then from 50 simulated populations that were either linear or curved by design. From each population we randomly drew 200

sets of neural population responses to between 3 and 7 unique options. Sets (or “trials”) always included two good offers (e.g. value bins 24 and 25 at the high end of the range [Figure 5] or value bins 12 and 14 [Supplementary Figure S5]) with the remaining offers chosen at random from the lower portion of the value range. To find the upper bound on choice decoding accuracy for each choice set, we used the decoding model described above. The “decoding accuracy” was defined as the percent of “trials” where the model correctly identified the best option as being more valuable than the 2nd best option, i.e. the likelihood that the decoded highest-value-option matched the objectively best option. Decoding accuracy was calculated within each population, for each choice set. The measured “decoy effect” for each population was then the effect of the worst option’s value on decoding accuracy (i.e. the slope of decoding accuracy by the minimum value in the set).”

The results are reported in the text as follows (p. 16, lines 442-458):

“We found that curvature in the value-encoding manifold was sufficient to produce strong decoy effects in decoding accuracy (Figure 5B-C; curved population, mean slope of the decoy effect = 0.17, range = [0.02, 0.40], $p < 0.0001$, one-sample t-test from 0). Linearized populations also produced a weaker but positive decoy effect (mean slope of the decoy effect = 0.05, range = [-0.13, 0.18], $p < 0.0001$), suggesting that the linearized population still maintained a level of curvature capable of affecting the readout, likely due to the Poisson spiking close to the noise floor (Okazawa et al. 2021). This result was not an artifact of considering some especially wide range of values: curvature in value-encoding manifold was sufficient to produce strong effects on decoding accuracy when we only considered a narrower range of values (Supplementary Figure S5). To test the role of curvature in producing the decoy effect we additionally simulated 50 curved and linear populations with slightly elevated baseline firing rates to remove them from the noise floor (Supplementary Figures S6, S7). Here, we found that curvature in the manifold was necessary to produce decoy effects in decoding accuracy, as the decoy effect was absent in simulated linear populations (curved population, mean slope of the decoy effect = 0.191, range = [0.009, 0.508], $p < 0.0001$; linear population, mean slope of the decoy effect = 0.005, range = [-0.079, 0.121], $p = 0.436$).”

Figure 5. Curvilinear manifolds suggest a specific pattern of irrational decisions. A) A cartoon illustrating how the ability to discriminate good offers could depend on the value of a worst, “decoy” offer in a curved manifold. The linear decoder represents an upper bound on the accuracy of a downstream neuron’s readout from vmPFC. The inset graph shows how the probability of choosing the best option changes as a function of decoy offer value in an example vmPFC pseudopopulation; the slope here is the “decoy effect”: the effect of the worst offer’s value on the accuracy of the choices (marked in [B] with an open arrow). **B)** Distributions of decoy effect slopes from vmPFC pseudopopulations (black) and their linearized version (purple). Filled arrows = means, open arrow = the example effect illustrated in (A). **C)** Same as (B) but the distributions of decoy effects were obtained from simulated populations that had either a curved (black) or linear (purple) value manifold.

Figure S5. Decoding from a constrained range of values in vmPFC (real and linearized), related to Figure 5. A) A cartoon illustrating how the ability to discriminate good offers could depend on how bad the worst offer is in a curved manifold, when the best values in the set are around the middle of the full range of the manifold and the decoding range is thus constrained (black horizontal braces). **B)** Distributions of decoy effects from vmPFC pseudopopulations (black) and their linearized version (purple) value manifold constrained to the lower 56% of values (value bins 1-14 from the full range of 25, see **Methods**, vmPFC population, mean decoy effect slope = 0.326, range = [-0.003, 0.588], $p < 0.0001$, linearized population, mean decoy effect slope = 0.111, range = [-0.103, 0.377], $p < 0.0001$). Filled arrows indicate the means of the distributions.

Another question about Figure 5 is why the worst offer value (E-F) stops at 0.5. Is it simply that there is not much data above this point because the animals always accept the initial stimulus?

Yes, there were few decoy offer values exceeding ~ 0.5 , either because these offers were unlikely to be the lowest offer in any given set or because offers that were above threshold were sometimes accepted without considering additional offers (**Supplementary Figure S10**). We have clarified the legend of **Figure 5** to indicate that decoy offers are quantile binned.

Figure S10. Errors and returns depending on the best, best-2nd best and worst offer values (for trials in which at least 3 offers were revealed), related to Figures 1 and 5. A) Distribution of trials in which the best offer was chosen (i.e. the choice was “correct”; gray) or some other offer was chosen (i.e. an “error” was made; red), depending on the value of the best, best-2nd best and worst offer revealed in the trial. **B)** Same as A, but for trials which contained only sequential reveals, (gray) versus those that contained returns (black), depending on the value of the best, best-2nd best and worst offer revealed in the trial.

Also, in these panels the difficulty of the best/2nd best discrimination is fixed by restricting to a 20% difference, but it appears that the data are collapsed over the range of best offer values. However, this seems like a relevant parameter for behavior. I.e., does the decoy effect vary depending on the best offer value (test by including this or average of best and 2nd best value in GLM)? The model in Fig. 5A suggests not, a decoy close in value to best and 2nd best offers that have high values should have the same improved discriminability as a decoy close in value to best and 2nd best offers with low values.

We thank the reviewer for prompting us to direct more attention to the relationship between trial difficulty and the effect of the worst offer. In place of analyzing only the “difficult trials”, as we had before, we now consider the full interaction between difficulty and decoy-offer value in our plots and computational models (**Figure 5**; **Supplementary Figure S8**; **Table S2-3**). These results only reinforce our original conclusions.

In order to explain the changes we have made to the manuscript, we should say that reviewer #1’s major comment 1 prompted us to reconsider our understanding of how the subjects solved the task. In short, the revised manuscript reports that the subjects were using a hybrid strategy in which they compared values against a threshold, but also compared values against each other. This means that trials could be difficult either because the best value was close to threshold or because the difference between the best and 2nd-best option were close together. We find that both of these forms of difficulty affected the probability of making an error, but also a novel measure of uncertainty developed in the revised manuscript: the animals’ tendency to return to re-evaluate previously viewed options (revised **Figure 5**; new **Supplementary Figures S8 & S10**). Critically, both forms of difficulty also interacted with the decoy effect, meaning that the decoy effect was most pronounced for the most difficult trials (**Table S2-3**).

Because we discovered in addressing this question that most errors (and to a lesser extent returns) happened when the best values were around the middle of the value range (**Supplementary Figure S10**), we also took the additional step of verifying that the manifold curvature also predicts irrational decisions within this restricted range of values. Indeed it does (**Supplementary Figure S5**): we obtained a similar, strong decoy effect in decoding choice accuracy even within the restricted range of values that most closely matched the real distribution of error and return trials.

The revised manuscript now describes the subjects’ strategies in the task as follows (p. 6, lines 145-163):

“Because the subjects could return to a previously viewed offer, their strategy in this task was best described as a mixture of 2 kinds of decision-making processes: a compare-to-threshold process (where each offer was sequentially evaluated against a threshold and accepted immediately if it was clearly above threshold) and a value-comparison process (where newly encountered offers were compared against previously-viewed and still-available offers). In a compare-to-threshold process, the decision is most difficult if values are not clearly above or below the threshold, in particular, if the best option in forward sweep through the offers is still relatively low. In a value-comparison process, the decision is most difficult when good options are close in value and thus harder to discriminate. We found that the subjects were most likely to make errors (i.e. not choose the best offer) when either form of difficulty was high (**Figure 1K**; best offer value effect, subject J: 2.17, range = [1.17, 3.23], subject T: 2.65, range = [0, 3.81], both $p < 0.0001$, logistic regression; best-2nd best offer value effect, subject J: 2.11, range = [1.12, 3.09], subject T: 2.53, range = [0, 3.58], both $p < 0.0001$). Both subjects were also most likely to return to previously viewed offers when either form of difficulty was high (**Figure 1L**; best value, subject J: -1.65, range = [-3.28 - 0.04], subject T: -1.94, range = [-3.25, -0.65], both $p < 0.0001$; best-2nd best, subject J: -1.33, range = [-2.60, 0.95], subject T: -1.24, range = [-2.46, -0.12], both $p < 0.0001$).”

Figure 1. Task design and behavior. **D)** Probability of choosing an offer as a function of offer value. Gray line = logistic fit. Each dot represents an offer value. **E)** Distribution of the number of offers viewed per trial. Cyan line = prediction from an optimal compare-to-threshold model. Black line = best fit compare-to-threshold model (maximum likelihood). **F)** Probability of returning to a previously seen offer. **G)** Expected value of trials which did (“ret.” = return) or did not (“seq.” = sequential) contain a return to a previous offer. **H)** Average value of offers which were returned to (“ret.” = return) versus those which were seen only once (“seq.” = sequential). **I)** Probability of a given number of returns in a trial, in trials where multiple unique offers were revealed (>3). **J)** Probability of the return in multiple offer trials as a function of the viewing time at which an offer was finally accepted (at position 0). **K)** Probability of choosing the best option as a function of value of the best offer revealed within a trial (left) or as a function of the difference in value between the best and the 2nd best offer(right), multiple offer trials. **L)** The same as (K) but for returning to a previously seen option. Lines = least squares fit. Error bars indicate \pm standard error of the mean across sessions (SEM).

The GLM with interaction terms is described in the Methods (p. 28, lines 913-926):

“To determine whether the value of the irrelevant decoy option affected the probability of choosing the best option, we fit a GLM that included the best option’s value, the difference between the best and second-best options and the main effects of the decoy option’s value, together with its pairwise interactions with the two other factors:

$$\log \left(\frac{p(\text{chose best})}{1 - p(\text{chose best})} \right) = \beta_0 + \beta_1 V_{\text{best}} + \beta_2 (V_{\text{best}} - V_{2\text{nd}}) + \beta_3 V_{\text{decoy}} + \beta_4 (V_{\text{best}} \times V_{\text{decoy}}) + \beta_5 ((V_{\text{best}} - V_{2\text{nd}}) \times V_{\text{decoy}})$$

Where V_{best} is the value of the best option, $V_{2\text{nd}}$ is the value of the second-best option, and V_{decoy} is the value of the decoy option. To ensure that potential errors in choosing the best option are made among the good options, and not caused by choosing the decoy, we excluded trials in which subjects chose the decoy option ($n = 472$, 3%). The model was fit with L2 regularization ($\lambda = 1$) to ensure the stability of parameter estimates. Analogous analyses were also performed for probability of making a return within a trial, i.e. making a repeated reveal of a previously seen option.”

The relationship between the decoy effect and trial difficulty is reported as follows (p. 16-17, lines 465-495):

“In addition to being better at choosing the best option as it got larger with respect to the threshold and/or in comparison to the second best option, the subjects were also better at choosing the best option as the value of the worst, decoy option in each set got higher (**Figure 5D-E**; both subjects: mean decoy effect slope = 0.17, range = [-0.56, 0.78], $p < 0.0001$; GLM included terms for choosing the best option by best value, best-2nd best value, decoy value, and pairwise interactions with decoy value, similar results for individual

subjects and different parameterizations of the GLM, **Supplementary Tables S2 and S3**). While accuracy for very high best offer values approached ceiling, a strong effect of decoy value was observed for lower best offer values and best offer values that were close to the 2nd best offers (**Figure 5F**; both subjects: decoy by best interaction, mean slope = 0.27, range = [0.13, 0.43], $p < 0.0001$; decoy by best-2nd best interaction, mean slope = 0.35, range = [-0.03, 0.67], $p < 0.0001$; similar results for individual subjects, **Supplementary Table S2, Supplementary Figure S8A-C**).

Errors of reward maximization are only one of the consequences of uncertainty about the best option in the task. As described in **Figure 1**, the monkeys were also more likely to return to previous options on trials where the best decision was uncertain. Therefore, we reasoned that any effect of decoys on the discriminability of good options or of good options from threshold might also result in an increase in returns. As predicted, subjects were less likely to dither between options (i.e. they made fewer returns) when the value of the worst option in the set got higher (**Figure 5G-I**). This was true even after taking into account the best offer value and the difference between the best and 2nd best offers (both subjects: mean decoy effect slope = -0.45, range = [-1.24, 0.52], $p < 0.0001$; same GLM as for accuracy, similar results for individual subjects, **Supplementary Table S2**). Again, the decoy effect was most pronounced for lower best offer values and best offer values close to the 2nd best offers (**Figure 5I**; both subjects: decoy by best interaction, mean slope = -0.43, range = [-1.01, 0.48], $p < 0.0001$; decoy by best-2nd best interaction, mean slope = -0.18, range = [-0.39, 0.10], $p < 0.0001$; similar results for individual subjects, **Supplementary Table S2; Supplementary Figure S8D-F**). Together, these results converge to suggest that the subjects were more uncertain about which offer to choose when irrelevant decoy offer values were lower, tracking the upper bound on decoding accuracy predicted by the curved manifold.”

Figure 5. Curvilinear manifolds suggest a specific pattern of irrational decisions. D) Probability of choosing the best offer seen as a function of the worst offer seen (decoy offer quantile). Lines = least squares fit. **E)** Distribution of decoy effect slopes across sessions. **F)** Probability of choosing the best option as a function of the decoy value and other variables that affect accuracy: Left) The best value seen. Right) The difference between the best and 2nd best option. **G-I)** Same as (D-F), but for the probability of returning to a previously seen offer. Insets in **G)** show which options tended to be the target of returns. Error bars indicate \pm standard error across sessions (SEM). These may be smaller than the symbols. **** $p < 0.0001$.

Figure S8. Decoy effect within most confusable trials, related to Figure 5. A) Probability of choosing the best offer seen as a function of the worst offer seen within trials, in which the best and 2nd best offer values were < 0.2 from each other (n , subject J = 2,937; n , subject T = 2,866). Line = least squares fit. **B)** Distribution of decoy effects within the probability of choosing the best option in the set across sessions. **C)** Probability of choosing the best option seen within each trial as a function of both the best value and the decoy value in the set (left) and the difference between the best and 2nd best option, and the decoy value (right). **D-F)** Same as (A-C) but for the probability of return. **** $p < 0.0001$.

Decoding from the constrained range of values is described as follows:

Methods (p. 27, lines 885-889): “From each population we randomly drew 200 sets of neural population responses to between 3 and 7 unique options. Sets (or “trials”) always included two good offers (e.g. value bins 24 and 25 at the high end of the range [Figure 5] or value bins 12 and 14 [Supplementary Figure S5]) with the remaining offers chosen at random from the lower portion of the value range.”

Results (p. 16, lines 449-451): “This result was not an artifact of considering some especially wide range of values: curvature in value-encoding manifold was sufficient to produce strong effects on decoding accuracy when we only considered a narrower range of values (Supplementary Figure S5).”

Minor

Although monotonicity is evoked in the Results when referring to the average response across all neurons (L. 152), the authors state that this is a linear relationship in the Discussion (L. 432), contrasting this with the nonlinearity observed in some individual neurons. Linearity was not actually tested for the average, which doesn't actually look linear, so this sentence should be modified to reflect better the figure.

Thank you for spotting that - it has been corrected. We also added a curved fit to Figure 2E for reference/comparison.

Figure 2. Tuning for value in individual vmPFC neurons. E) Average firing rates from all 122 neurons, plotted as a function of value quantile.

Do "offer size" in Fig. 1 and "offer value" in Fig. 5 index the same measure, the latter being divided by 100? If so, maybe make the language and scale consistent.

We apologize for the inconsistency. We now talk about "offer values", using the same scale, throughout the manuscript.

In some of the tables the number of samples is listed. Would be nice if this was done for all the model summaries. Also, it is unclear where n=5803 in Table S5 comes from?

This was the number of trials which had at least 3 offers revealed and the difference between the best and second-best option within 0.2 of the value range (now only relevant to **Supplemental Figure S8** and provided in the methods and figure legend). We now conduct main analyses without sub-selecting most confusable trials and include the number of observations in each table.

The modified description in the Methods now reads as follows (p. 28, lines 907-911):

"we additionally repeated our analyses on the trials where the difference between the best and second-best option were within 0.2 (**Supplementary Figure S8**; 8,157 total trials; subject J: 4,037; subject T: 4,120; 5,803 trials which had at least 3 options revealed, subject J: 2,937; subject T: 2,866)."

Reviewers' Comments:

Reviewer #1:

Remarks to the Author:

I thank the authors for fully addressing my concerns. I have no further comments. Congratulations on the excellent work relating neural geometry and irrational behavior in value decision making.

Reviewer #2:

Remarks to the Author:

I have read the authors' responses and the revised manuscript, and my main concern remains unresolved. The concern is that there is a possibility that a neuron's response to an offer could be affected by the value of other offers viewed on that trial. Because there are uneven distributions of value differences between offers in this task, such effects could underlie the curvature they describe in the value code. The authors have dismissed this as a "complex and speculative explanation", which it is not. Neurons in the vmPFC have been shown to encode value comparisons, meaning that options besides the one currently being considered could influence neuron firing rates, and neurons in nearby OFC regions have been shown to adapt their responses based on the range of values available. The authors noted that they did simulations to determine that range adaptation is not sufficient to produce their results, however (A) either they do not show or describe these simulations or the simulations they're referring to do not actually address this problem, meaning there is no evidence presented to support their claim and (B) there is no need for simulations, which necessarily make assumptions, when they can use their empirical data to test the possibility. Therefore, I am not convinced that this concern should be dismissed based on the evidence presented.

The authors argue that the u-shaped "tuning functions" of single neurons create the curved manifold that they describe. I agree. The point of my comment was that those shapes – in the single neurons and also in the population response – may be created by systematic differences in the trials that contribute to each condition. The systematic difference is that the options on the end of the value spectrum have the potential to be seen in the same trial with options that have a large difference in value, whereas the options in the middle of the value spectrum can only be seen with options that are somewhat different in value. There is a very straight-forward way to remedy this problem, which is to restrict the analyses to only the first "option viewing epoch" on each trial, and remove any sampling of additional options. When viewing the first option, the animal would not know the values of other offers, so any effects other options might have on neuron responses would be removed. The animals performed something over 20,000 trials total, so there should be more than enough data to do this. Moreover, since the options are masked, the first option viewed should have a random value, so the value conditions in this analysis should be better balanced than when subsequent offers are included. Can the authors please show that their results in Figures 2, 3, and 4 hold up when only the first option viewing is considered.

Other comments

The first and second panels of the new Figures 1K and 1L are likely showing highly correlated data. This is because the 'best – second best value' measure can only be large when the best value is high. If the authors want to show separate effects for each measure, they should check whether this is true and control for the correlation as needed.

The data in the new Figure S9A appears to be inconsistent with the data in Figure 1E. For example, 3 offers viewed on 50% of trials in S9A versus 0.15 probability of viewing 3 offers in 1E.

In Figure 1G, it should be clarified how expected value is calculated. Is it based on the highest option viewed, or some combination of options viewed? If the former, why not call it 'highest option value' to

avoid confusion?

The sentence in lines 577-579 is a fragment.

Reviewer #3:

Remarks to the Author:

The authors did a great job revising the manuscript to address my concerns. I just have a follow-up related to the new supplementary analyses (related to Figure S3 & S4). I appreciated seeing the top 10 cells with the highest absolute PC2 loadings (green). From Figure S4 it seems that excluding 2 of these 10 cells is sufficient to abolish curvature along PC2. Can the authors indicate in Figure S3 which cells correspond to which rank in Figure S4A? Are the 2 cells driving PC2 curvature the non-monotonic cells in Figure S3 tuned for specific values? It would be nice to see how these cells load onto PC1 and PC3 as well. If I understand the pseudopopulation algorithm, all cells recorded were sampled in each pseudopopulation, which would not capture the effect of a small subset of cells driving for example the decoy effect. I assume that excluding the 2 cells driving PC2 curvature would abolish the decoy effect?

Minor

line 40, Figure S3 - It is mentioned that only 19% of tuned neurons were better described using the piecewise linear compared to the curvilinear fit. How was this model comparison made? For what percentage of neurons was the curvilinear fit better compared to piecewise linear? What percentage were they equivalent (assuming your comparison method allows this)?

Reviewer #1:

I thank the authors for fully addressing my concerns. I have no further comments. Congratulations on the excellent work relating neural geometry and irrational behavior in value decision making.

We appreciate the reviewer's kind words and endorsement of publication.

Remarks on code availability:

All the main figures are reproducible with the available code and data.

Reviewer #2:

I have read the authors' responses and the revised manuscript, and my main concern remains unresolved. The concern is that there is a possibility that a neuron's response to an offer could be affected by the value of other offers viewed on that trial. Because there are uneven distributions of value differences between offers in this task, such effects could underlie the curvature they describe in the value code. The authors have dismissed this as a "complex and speculative explanation", which it is not. Neurons in the vmPFC have been shown to encode value comparisons, meaning that options besides the one currently being considered could influence neuron firing rates, and neurons in nearby OFC regions have been shown to adapt their responses based on the range of values available. The authors noted that they did simulations to determine that range adaptation is not sufficient to produce their results, however (A) either they do not show or describe these simulations or the simulations they're referring to do not actually address this problem, meaning there is no evidence presented to support their claim and (B) there is no need for simulations, which necessarily make assumptions, when they can use their empirical data to test the possibility. Therefore, I am not convinced that this concern should be dismissed based on the evidence presented.

The authors argue that the u-shaped "tuning functions" of single neurons create the curved manifold that they describe. I agree. The point of my comment was that those shapes – in the single neurons and also in the population response - may be created by systematic differences in the trials that contribute to each condition. The systematic difference is that the options on the end of the value spectrum have the potential to be seen in the same trial with options that have a large difference in value, whereas the options in the middle of the value spectrum can only be seen with options that are somewhat different in value. There is a very straight-forward way to remedy this problem, which is to restrict the analyses to only the first "option viewing epoch" on each trial, and remove any sampling of additional options. When viewing the first option, the animal would not know the values of other offers, so any effects other options might have on neuron responses would be removed. The animals performed something over 20,000 trials total, so there should be more than enough data to do this. Moreover, since the options are masked, the first option viewed should have a random

value, so the value conditions in this analysis should be better balanced than when subsequent offers are included. Can the authors please show that their results in Figures 2, 3, and 4 hold up when only the first option viewing is considered.

We appreciate the reviewer clarifying what analyses would address these concerns. The revised manuscript now shows that the results in Figures 2, 3, and 4 do indeed hold up when only the first option viewing is considered. These new results are included in **Supplementary Figure S4**. They reinforce the conclusion that curvature in single neuron and population responses could not stem from any kind of sequence effects, value comparison process, range adaptation, or other contextual process because all of the key results are still apparent when we only consider the first offer period. Including these analyses in the manuscript substantially strengthens the evidence for our central conclusions.

Figure S4. Representational geometry of value based on first offers revealed, related to Figure 2, 3 and 4. **A)** The firing rates of example neurons that were quadratically (left column) or non-quadratically (middle column) tuned for value or not tuned for value (right column), plotted as a function of value quantile bins. **B)** Proportion of all cells ($n = 122$) within each category. **C)** Average firing rates from all 122 neurons, plotted as a function of value quantile. Error bars indicate \pm standard error of the mean across neurons (SEM). **D)** The projection of the neural population onto the first 2 principal components (PCs). Shades of gray = value bins from low (light gray) to

high (dark gray). Dotted line = best linear fit. Solid line = best quadratic fit. **E)** Percent variance explained by each PC. **F)** A comparison of the variance explained by the first 2 PCs in the real population (vertical line) against bootstrapped distributions of linearized datasets. **G)** A linear decoder trained on the vmPFC population and used to predict value. **H)** Decoders trained on the population response to one half of the values (filled circles) and used to predict values outside of this range (open circles). Red = trained on high values; blue = trained on low values. **I)** The mean distance between neuronal states corresponding to different values. **** $p < 0.0001$.

These results are referenced in the text as follows (p. 8, lines 201-203 and p.18, lines 521-524):

“The following results were also not due to viewing offers sequentially because all results were also observed just within the first offer viewing period of each trial (**Supplementary Figure S4**).”

“Again, these effects did not appear to be the simple consequence of divisive normalization. The curvature in single-cell and population data is similar when we only consider the first offer viewing period, when the context of other offers in the trial is not yet present (**Supplementary Figure S4**).”

Other comments

The first and second panels of the new Figures 1K and 1L are likely showing highly correlated data. This is because the ‘best – second best value’ measure can only be large when the best value is high. If the authors want to show separate effects for each measure, they should check whether this is true and control for the correlation as needed.

It is important to be clear that none of the analyses or conclusions of the paper depend on the notion that these measures independently predict the behavior. That said, we think it’s a great idea to add an illustration to show the relative strength of each measure, controlling for the other. This new analysis is now **Supplementary Figure S2**. The statistical approach in the last version of the manuscript already included both factors to ensure that variability in one factor was not mis-ascribed to the other. Although the procedure we used (multiple regression) is the more accurate way to control for correlated variables (because it yields unbiased parameter estimates; e.g. Freckleton, 2002) we believe that the new regression on the residuals analysis does give some valuable intuition about the marginal magnitude of each effect. The supplemental figure legend reports that both best value and best-2nd best value predicted correct choice and returning behavior even when any contribution of the competing factor was first regressed out of the behavior.

“Figure S2. Effects of best offer and best-2nd best offer values on residual behavioral performance, related to Figure 1. A) Probability of choosing the best option as a function of best offer value after regressing out best-2nd best offer value (left, both subjects: 2.84, range = [1.22, 5.25], $p < 0.0001$, subject J: 2.40, range = [1.22, 3.75], subject T: 3.24, range = [0, 5.25], both $p < 0.0001$), and as a function of best-2nd best offer value after regressing out best offer value (right, both subjects: 2.64, range = [1.14, 4.61], $p < 0.0001$, subject J: 2.31, range = [1.14, 3.64], subject T: 2.93, range = [0, 4.61], both $p < 0.0001$), multiple offer trials. **B)** The same as (A) but for returning to a previously seen option (returning to a previous option as a function of best offer value after regressing out best-2nd best offer value, both subjects: -2.27, range = [-5.29, -0.41], $p < 0.0001$, subject J: -2.08, range = [-5.29, -0.41], subject T: -2.47, range = [-4.76, -0.91], both $p < 0.0001$); returning to a previous option as a function of best-2nd best offer value after regressing out best offer value, both subjects: -1.28, range = [-2.86, 1.93], $p < 0.0001$, subject J: -1.38, range = [-2.86, 1.93], subject T: -1.18, range = [-2.53, 0.16], both $p < 0.0001$). Lines = least squares fit. Error bars indicate \pm standard error of the mean across sessions (SEM).”

We added explanation in the figure caption:

“Figure 1. K) Probability of choosing the best option as a function of value of the best offer revealed within a trial (left) or as a function of the difference in value between the best and the 2nd best offer (right), the two collinear factors influencing overall difficulty of the trial, multiple offer trials, see also **Supplementary Figure S2A. L)** The same as (K) but for returning to a previously seen option, see also **Supplementary Figure S2B.**”

The text of the results was modified as follows (p. 6, lines 150-166):

“In a compare-to-threshold process, the decision is most difficult if values are not clearly above or below the threshold, in particular, if the best option in forward sweep through the offers is still relatively low. In a value-comparison process, the decision is most difficult when good options are close in value and thus harder to discriminate. We found that the subjects were most likely to make errors (i.e. not choose the best offer) when difficulty, understood as either low best value and/or small difference between the best options, was high (**Figure 1K**; best offer value effect, subject J: 2.17, range = [1.17, 3.23], subject T: 2.65, range = [0, 3.81], both $p < 0.0001$, logistic regression; best-2nd best offer value effect, subject J: 2.11, range = [1.12, 3.09], subject T: 2.53, range = [0, 3.58], both $p < 0.0001$). Both subjects were also most likely to return to previously viewed offers when the difficulty was high (**Figure 1L**; best value, subject J: -1.65, range = [-3.28 - 0.04], subject T: -1.94, range = [-3.25, -0.65], both $p < 0.0001$; best-2nd best, subject J: -1.33, range = [-2.60, 0.95], subject T: -1.24, range = [-2.46, -0.12], both $p < 0.0001$). Because the two types of difficulty tended to co-occur (option values tended to be closer when no option was high; $r = 0.72$; **Supplementary Figure S12**), we further verified that both forms of difficulty predicted both errors and returns via regression on the residuals from models that controlled for the other type of difficulty (**Supplementary Figure S2**).”

The data in the new Figure S9A appears to be inconsistent with the data in Figure 1E. For example, 3 offers viewed on 50% of trials in S9A versus 0.15 probability of viewing 3 offers in 1E.

We appreciate the reviewer spotting this issue. The header in former Figure S9A—now **Figure S11A**—incorrectly suggested that these plots were of the same subsets of trials. Figure 1E shows proportion per all trials, while Figure S11A shows proportion per 3-or-more offer trials because only these were used for calculating decoy effects. We have corrected the error in the header of Figure S11A.

In Figure 1G, it should be clarified how expected value is calculated. Is it based on the highest option viewed, or some combination of options viewed? If the former, why not call it 'highest option value' to avoid confusion?

Thank you for pointing out that “expected value” was not defined in the text. We used the term to mean the average of revealed offers.

We amended the results to define the term (p. 5, line 129):

“Average value of revealed offers (trial expected value, EV)“

The sentence in lines 577-579 is a fragment.

We reformulated the sentence (p. 19, lines 576-579):

“The links we draw between the curved value-coding manifold and irrational decision-making may seem predicated on some provocative premises. The first of these is the notion that a downstream region would decode value in a way that is essentially linear. There is precedent for this view.”

Remarks on code availability:

the code doesn't run to completion. there are some functions and variables missing - nanste.m, errbar.m, bootstrapExplainedVariance_vNon.m, pop.mat, and the code runs into errors after excluding these lines too.

We apologize for omitting these functions. We added all the missing, supporting functions and tested the code on a different computer to ensure clear path register. The code was tested in Matlab 2020a. The new version of the code is available in the same directory https://osf.io/8mkrd/?view_only=9a491e6e508f4532bff15d1032cabd90

Reviewer #3:

The authors did a great job revising the manuscript to address my concerns. I just have a follow-up related to the new supplementary analyses (related to Figure S3 & S4). I appreciated seeing the top 10 cells with the highest absolute PC2 loadings (green). From

Figure S4 it seems that excluding 2 of these 10 cells is sufficient to abolish curvature along PC2. Can the authors indicate in Figure S3 which cells correspond to which rank in Figure S4A? Are the 2 cells driving PC2 curvature the non-monotonic cells in Figure S3 tuned for specific values? It would be nice to see how these cells load onto PC1 and PC3 as well.

We appreciate the reviewer's kind words about our efforts in the last round of review and hope that the following changes satisfy all remaining concerns.

First, to better illustrate the relationship between individual cells and PCs, we now color-code the rank along absolute loadings on PC2 for all cells in **Figure S5** and **S6** (former Figures S3 and S4). Additionally, we plot all the tuned cells with both quadratic and piecewise linear fits ranked by the contribution to PC2. The top first and second cells are non-monotonic and monotonic, respectively.

“Figure S6. A) PC loadings for PC1, PC2 and PC3 across the neurons sorted from highest to lowest. Cells are color-coded by their rank on PC2.”

“Figure S5. B) Each point represents one neuron that was either significantly tuned (filled circles) or untuned (open circles) to the offer value according to the test based on mutual information between firing rates and value labels (see Methods). Shading indicates absolute PC2 loadings, with the darkest green for cells with the highest contributions (see Supplementary Figure S6). The piecewise linear model fits independent slopes to neural responses to high and low values, with the break point constrained to fall between the 20% lowest and the 20% highest values. Points along the vertical axis only have a slope for high values, points along the horizontal axis only have a slope for low values. There was an overall negative correlation between lower and higher value slopes ($r = -0.28$, $p < 0.002$), supporting the finding of non-linear shapes in single cell responses. Less than 20% [9/46] of the tuned neurons were better described with piecewise linear fit than the curvilinear fit (Mandel’s test, see Methods) suggesting that floor (downward rectified) or ceiling (upwards rectified) effects were not a better explanation for the majority of the tuned responses. Quadratic fit was better than piecewise linear fit in ~13% [6/46] of tuned cells, leaving ~67% [31/46] of them not better described by either piecewise linear or quadratic fit. C) The firing rates of all non-linearly tuned neurons, plotted with quadratic (gray line) and piecewise linear (pink line) fits. The green boxes with a number indicate the cell’s rank along PC2 absolute loadings (see Supplementary Figure S6).”

If I understand the pseudopopulation algorithm, all cells recorded were sampled in each pseudopopulation, which would not capture the effect of a small subset of cells driving for example the decoy effect. I assume that excluding the 2 cells driving PC2 curvature would abolish the decoy effect?

Excluding the cells with the highest PC2 loadings does not abolish the decoy effect, as we now show in **Figure S6D**. It is also important to clarify that more than 2 cells are driving the curvature in the population. We suspect that the original panel headers in **Figure S6** may have caused some confusion, so we have re-written them to be less ambiguous. Hopefully this change clarifies that the first column (with clear curvature) is after dropping the top 2 cells, the second column (also with clear curvature) is after dropping top 4 cells, etc. Further, revised **Figure S6** shows that dropping >10 of the PC2 cells does not disrupt the curvature in higher dimensions nor abolish the decoy effect. We are thus confident that these results are not an artifact of 2 unusual cells.

“Figure S6. C) The projection of the neural population onto PC1 and PC2 or PC1 and PC3 after excluding 2, 4, 6, 8 or 10 of the neurons with highest weights in the PC2 (see also **Supplementary Figure S5** for piecewise linear regression weights for these neurons). Excluding the neurons with highest PC2 loadings removes the curvature from the PC1 vs PC2 projection. However, curvature can still be found in PC3. Shades of gray = value bins from low (light gray) to high (dark gray). Dotted line = best linear fit. Solid line = best quadratic fit. **D)** Distributions of decoy effect slopes from vmPFC pseudopopulations after excluding 2, 4, 6, 8 or 10 of the neurons with highest weights in the PC2 (black) and their linearized version (purple). Filled arrows = means of the distribution. * $p < 0.01$, ** $p < 0.001$, **** $p < 0.0001$.”

Minor

line 40, Figure S3 - It is mentioned that only 19% of tuned neurons were better described using the piecewise linear compared to the curvilinear fit. How was this model comparison made?

Apologies for the omission. We used the Mandel's fitting test, the same method that was earlier applied for determining the nonlinearity of single neuron responses (i.e. better quadratic fit than linear).

Information on the method of comparison was added in the Methods (p. 25, lines 815-817):
"These results are included in **Supplementary Figure S5**. We repeated Mandel's fitting test to compare piecewise linear and quadratic fits."

For what percentage of neurons was the curvilinear fit better compared to piecewise linear? What percentage were they equivalent (assuming your comparison method allows this)?

Thanks for catching this issue. We have revised the manuscript to report these results and avoid making strong claims about the curvilinear fit being better than the piecewise linear fit: both are perfectly capable of capturing the non-monotonic tuning functions we observed here.

The revised text is included in Supplementary Figure S5:

*"**Figure S5. B**) (...) Less than 20% [9/46] of the tuned neurons were better described with piecewise linear fit than the curvilinear fit (Mandel's test, see **Methods**) suggesting that floor (downward rectified) or ceiling (upwards rectified) effects were not a better explanation for the majority of the tuned responses. Quadratic fit was better than piecewise linear fit in ~13% [6/46] of tuned cells, leaving ~67% [31/46] of them not better described by either piecewise linear or quadratic fit."*

Remarks on code availability:

I could see the filenames of the code and data, but I was unable to download for review.

Unfortunately, the OSF website is not as user friendly as it could be. To download the files, after clicking on the folder with code or on the data file, it is necessary to find the "three dots" button to the right of the file name, and select the download option there. The new version of the code is available in the same directory

https://osf.io/8mkrd/?view_only=9a491e6e508f4532bff15d1032cabd90

Reviewers' Comments:

Reviewer #2:

Remarks to the Author:

The authors' additional analyses have adequately addressed my questions and improved confidence in the veracity of their findings. I have no further concerns.

Reviewer #3:

Remarks to the Author:

I thank the authors for their thorough responses to my comments/concerns. I have no further questions, and feel that the revised manuscript is appropriate for acceptance.